# Herbal Combinational Medication of *Glycyrrhiza glabra*, *Agastache rugosa* Containing Glycyrrhizic Acid, Tilianin Inhibits Neutrophilic Lung Inflammation by Affecting CXCL2, Interleukin-17/STAT3 Signal Pathways in a Murine Model of COPD

**DOI:** 10.3390/nu12040926

**Published:** 2020-03-27

**Authors:** Seung-Hyung Kim, Jung-Hee Hong, Won-Kyung Yang, Jeong-Ho Geum, Hye-Rim Kim, Su-Young Choi, Yun-Mi Kang, Hyo-Jin An, Young-Cheol Lee

**Affiliations:** 1Institute of Traditional Medicine & Bioscience, Daejeon University, Daejeon 34520, Korea; sksh518@dju.ac.kr (S.-H.K.); ywks1220@dju.ac.kr (W.-K.Y.); 2Department of Herbology, College of Korean Medicine, Sangji University, 83 Sangjidae-gil, Wonju, Gangwon-do 26339, Korea; anifam@hanmail.net; 3COSMAX NBT, INC., Seoul 06132, Korea; geumjeongho@cosmaxnbt.com (J.-H.G.); sychoifwp@cosmaxnbt.com (S.-Y.C.); 4COSMAX NS, INC., Seoul 06132, Korea; hyerimkim@cosmaxnbt.com; 5Department of Pharmacology, College of Korean Medicine, Sangji University, 83 Sangjidae-gil, Wonju, Gangwon-do 26339, Korea; yunmi6115@naver.com (Y.-M.K.); hjan@sj.ac.kr (H.-J.A.)

**Keywords:** *Glycyrrhiza glabra*, *Agastacha rugosa*, COPD, CXCL-2, IL-17, STAT3

## Abstract

Chronic obstructive pulmonary disease (COPD) is caused by exposure to toxic particles, such as coal fly ash (CFA), diesel-exhaust particle (DEP), and cigarette smoke (CS), leading to chronic bronchitis, mucus production, and a subsequent lung dysfunction. This study, using a mouse model of COPD, aimed to evaluate the effect of herbal combinational medication of *Glycyrrhiza glabra* (GG), *Agastache rugosa* (AR) containing glycyrrhizic acid (GA), and tilianin (TN) as active ingredients. GA, a major active component of GG, possesses a range of pharmacological and biological activities including anti-inflammatory, anti-allergic, anti-oxidative. TN is a major flavonoid that is present in AR. It has been reported to have anti-inflammatory effects of potential utility as an anti-COPD agent. The COPD in the mice model was induced by a challenge with CFA and DEP. BALB/c mice received CFA and DEP alternately three times for 2 weeks to induce COPD. The herbal mixture of GG, AR, and TN significantly decreased the number of neutrophils in the lungs and bronchoalveolar lavage (BAL) fluid. It also significantly reduced the production of C-X-C motif chemokine ligand 2 (CXCL-2), IL-17A, CXCL-1, TNF-α, symmetric dimethylarginine (SDMA) in BALF and CXCL-2, IL-17A, CXCL-1, MUC5AC, transient receptor potential vanilloid-1 (TRPV1), IL-6, COX-2, NOS-II, and TNF-α mRNA expression in the lung tissue. Notably, a combination of GG and AR was more effective at regulating such therapeutic targets than GG or AR alone. The histolopathological lung injury was alleviated by treatment with the herbal mixture and their active ingredients (especially TN). In this study, the herbal combinational mixture more effectively inhibited neutrophilic airway inflammation by regulating the expression of inflammatory cytokines and CXCL-2 by blocking the IL-17/STAT3 pathway. Therefore, a herbal mixture of GG and AR may be a potential therapeutic agent to treat COPD.

## 1. Introduction

Radix Glycyrrhizae (Liquorice), one of the most-commonly used traditional medicines in many countries is the root of *Glycyrrhiza glabra* L. (GG), or *Glycyrrhiza uralensis* Fisch. or *Glycyrrhiza inflata* Bat. [1]. Traditionally, GG is primarily effective for moistening the lung to arrest cough, fatigue, asthma with cough, excessive phlegm, and for removing drug toxins [2]. Glycyrrhizic acid (GA) and flavonoids, the major components of the GG, have been used extensively as complementary or alternative medicines in asthma because of their several pharmacological effects including anti-asthmatic [3], anti-allergic [4], anti-inflammatory, and anti-oxidative activities [5]. Previous reports showed that chemical ingredients of *Agastache rugosa* (AR) contained mainly essential oils, flavonoids, and other constituents [6,7], and possess anti-oxidative and anti-inflammatory activities [8]. Tilianin (TN) is a major constituent of AR and has been found to exhibit a number of pharmacologic effects such as anti-inflammatory and antioxidant activities [9,10] and could serve as a new potential anti-COPD agent. Chronic obstructive pulmonary disease (COPD) is a major chronic lung disease characterized by lung inflammation and pathological changes including bronchitis, small airway remodeling, mucus production, and progressive and persistent lung dysfunction [11].

Inhaled coal fly ash (CFA) and diesel-exhaust particle (DEP) induce a broad inflammatory response. Neutrophils, epithelial cells, and macrophages are activated and produce excessive amounts of reactive oxygen species (ROS) and various inflammatory cytokines. Thus, stimulated epithelial cells and fibroblasts can produce elastic fibers and collagen that are crucial components of the extracellular matrix [12].

Neutrophilic inflammation is one of the most important characteristics of COPD; airway neutrophilia is associated with a lung dysfunction and leads to pulmonary fibrosis and emphysema [13]. CXCL-2 is a small secreted chemokine that belongs to the CXC chemokine family [14]; its critically involved in the migration of neutrophils into inflammatory tissues and binding to the chemokine receptor CXCR2. IL-17A binds to the IL-17RA receptor complex that leads to production of several inflammatory chemokines such as neutrophil chemoattractants (CXCL-1, CXCL-2) [15]. Further, IL-17A can regulate CXCL-2 expression through activating the IL-17A receptor signaling pathway in epithelial cells after neutrophils migrate to the inflammatory sites.

STAT3 activation by IL-6, IL-23, and tumor growth factor-beta (TGF-β) induces Th17 differentiation and IL-17 cytokine production. Moreover, STAT3 is coordinated with RORγt to stabilize Th17 cells and their function [16]. However, the exact mechanisms involved in COPD pathogenesis are not fully understood and therapeutic approaches can only aim at relieving symptoms.

Despite their numerous pharmacological activities including anti-inflammatory effects, the molecular mechanisms of action of the herbal combinational medication described above have been fully investigated in a COPD model. Herbal mixture and herb–herb interactions have recently become of interest; herbal combination therapies have been elucidated. Traditional herbal mixtures have come into focus since most herbal medicinal preparations are a combination of single herbal components.

Our preliminary studies showed that a mixture of GG and AR, in the ratio 1:4 by weight, demonstrated effective anti-inflammatory and anti-oxidant activities (data not shown). There are few reports describing the biological activity of GG, AR, and their mixtures in respiratory system diseases, and hence further studies are needed. In this study, we evaluated the immunotherapeutic potential of GG, AR, and their 1:4 ratio mixture in preventing response to CFA in a mice model of COPD. We examined whether a combination of GG and AR natural herbs might be more effective in regulating neutrophil infiltration and CXCL-2, IL-17/STAT3 expression through a synergistic effect.

We hypothesized that GG, AR, and their mixture might be able to limit CFA-induced inflammatory response in a mouse model of COPD and confirm the involvement of the CXCL-2, IL-17/STAT3 pathway in the observed therapeutic effects.

## 2. Materials and Methods

### 2.1. Preparation of GG, AR, and GG Plus AR Extract (Large Capacity Extraction)

*Glycyrrhiza glabra*, *Agastache rugosa*, their combinational medication, and the trial product were obtained from COSMAX NS INC. (Seoul, South Korea). Briefly, 100 kg *of Glycyrrhiza glabra*, 100 kg of *Agastache rugosa,* a mixture of 20 kg of *Glycyrrhiza glabra*, and 80 kg of *Agastache rugosa* were extracted at 70 °C for 4 h using 1500 kg of solvent consisting of 70% water and 30% ethanol. The extract was filtered and concentrated. Dextrin was added into the concentrate (dextrin: dry matter content = 3: 7) and GG, AR, and their mixture were dried with a yield of 25%, 13%, 16% respectively (extract powder g/raw material g). The trial product (GG + AR)* was composed of GG plus AR extract (62.5%), microcrystalline cellulose (30.75%), magnesium stearate (1.5%), silicon dioxide (1.5%), hydroxypropyl methylcellulose (HPMC, 1.23%), glycerin (0.12%), titanium dioxide (0.5%), and gardenia yellow (1.9%).

### 2.2. Chemicals and Reagents

Tilianin (purity ≥98) was purchased from Chengdu Biopurify Phytochemicals Ltd. (Sichuan, China). Glycyrrhizic acid (purity ≥97) was obtained from Sigma Aldrich (St. Louis, MO, USA). HPLC-grade water and acetonitrile were obtained from Honeywell Burdick & Jackson (Morristown, NJ, USA). Acetic acid and ammonium acetate were supplied by Daejung Chemical & Metals (Siheung, Korea) and Sigma Aldrich (St. Louis, MO, USA), respectively. Dimethyl sulfoxide was purchased from Samchun pure chemical (Seoul, Korea). The remaining materials were supplied as follows: Dulbecco’s modified Eagle’s medium (DMEM; Daegu, Korea), fetal bovine serum (FBS), Dulbecco’s phosphate buffered saline (DPBS; WelGene Co., Korea), streptomycin, and penicillin (Lonza, MD, USA), TRIZOL^®^ reagent (Invitrogen, Carlsbad, CA, USA), ammonium-chloride-potassium (ACK) lysing buffer (Gibco, Life Technologies Corporation, NY, USA), bovine serum albumin (BSA) (Thermofisher Scientific, Seoul, Korea), CXCL-1, CXCL-2, IL-17, MUC5AC, IL-6, TNF-α, iNOS, and TRPV1 primers from Bioneer (Bioneer, Daejeon, Korea). Diesel particulate matter (DEP, SRM 2975) and 3-(4,5-dimethylthiazol-2-yl)-2,5-diphenyltetrazoliumbromide (MTT) were purchased from Sigma-Aldrich (St. Louis, MO, USA). All other reagents and chemicals were obtained from Sigma-Aldrich. Coal fly ash (CFA) used contained the following phenolic compounds (μg/mg): hydroquinone (0.23), resorcinol (0.33), catechol (0.9), phenol (3.56), m + p cresol (7.55), o-cresol (4.42), making a total of 16.91; it further contained following aromatic amines (µg/mg): 1-naphthylamine (1.36), 2-naphthylamine (1.72), 3-aminobiphenyl (0.21), and 4-aminobiphenyl (0.16); and further, benzo-alpha-pyrene (83.39); and finally fly ash (μg/mg), SiO_2_ (4), Fe_2_O_3_ (2.3), Al_2_O_3_ (3.2), CaO (0.3), MgO (0.7), TiO_2_ (0.4), and ignition loss (0.4). The particles were generated from the complete-combustion product of 1 g coal at 700 °C as previously described [17].

### 2.3. Animal Model and Treatment Regimen

Male BALB/c mice (6–8 weeks old; weight 20–22 g) were obtained from The Jackson Laboratory (Bar Harbor, ME, USA). The mice were housed in a specific-pathogen-free barrier facility at 21 ± 2 °C with a relative humidity of 60% ± 10% under a 12 h light/dark cycle. Food and water were provided *ad libitum*. The animal protocol was approved by the committee for animal welfare at Daejeon University (DJUARB2019-026) on September 16, 2019. This study was performed in strict accordance with the Guide for the Care and Use of Laboratory Animals of the National Institute of Health, and in accordance with the guidelines of the Institutional Animal Care and Use Committee of South Korea, Research Institute of Bioscience and Biotechnology (Daejeon, Republic of Korea). Mice were divided into 11 treatment groups (*n* = 8 for each group): (a) BALB/c normal, (b) CFD-sensitized control mice, (c) positive control: 3 mg/kg dexamethasone-treated CFD-sensitized mice, (d) 100 mg/kg GG-treated CFD-sensitized mice, (e) 100 mg/kg AR-treated CFD-sensitized mice, (f) 50 mg/kg GG + AR-treated CFD-sensitized mice, (g) 100 mg/kg GG + AR-treated CFD-sensitized mice, (h) 200 mg/kg GG + AR-treated CFD-sensitized mice, (i) 100 mg/kg (GG + AR)*-treated CFD-sensitized mice, (j) 10 mg/kg GA-treated CFD-sensitized mice, and (k) 10 mg/kg tilianin (TN)-treated CFD-sensitized mice. Following the acclimatization period, all groups except group 1 were administered 100 μL of CFD (coal 5 mg/mL, fly ash 10 mg/mL, and diesel-exhaust particles (DEP) 5 mg/mL) in saline by intratracheal instillation thrice at 3-day intervals for 12 days using bronchial tubes. Dexamethasone, GG, AR, GG+AR, GA, and TN were orally administered daily for 11 days at the above-mentioned dosages. On day 12, all mice were euthanized and blood, bronchoalveolar lavage fluid (BALF), and lung tissues were collected for further experiments

### 2.4. Collection of Bronchoalveolar Lavage Fluid (BALF) and Lung Cells

Single cell suspensions from lung tissues were isolated by mechanical disruption individually. Briefly, the extracted right lungs were minced using sterile scalpels, followed by incubation in PBS containing 1 mg/mL collagenase IV and 2 mg/mL dispase for 40 min at 37 °C in a sterile polypropylene tube. After incubation, lung tissue was vigorously pipetted up and down, and then filtered using a 70 μm cell-strainer. The remaining lobe of the left lung was stored for RNA extraction (upper lobe) and histological analyses (bottom lobe). BALF was collected 24 h after the last oral injection of samples. Mice were anesthetized by an intraperitoneal (i.p.) injection of 10% urethane (100 µL; Sigma-Aldrich, Korea). A tracheotomy was then performed, and a cannula inserted into the trachea. Ice-cold DMEM was instilled into the lungs, and BALF collected. The total cell counts were measured with a hemocytometer. For cytological examination, cells were prepared with a Cytospin (Hanil Science, Korea), fixed, and stained with a modified Diff-Quick stain. Differential cell counts were determined using at least 500 cells on each cytospin slide. Blood was collected by cardiac puncture, allowed to clot, then centrifuged; aliquots of serum were stored at ‒70 °C for ELISA.

### 2.5. Flow Cytometric Analysis

For two-color (double staining) fluorescence-activated cell sorting (FACS) analysis of lung tissues and BAL cells, enzymatic digestion of the lungs was performed. Briefly, mice were anesthetized, and the lungs carefully removed. After three washes, the lung tissue was cut into small pieces and transferred to a 15 mL conical tube containing 20 mL of HBSS with 2% FBS (Gibco-BRL, Grand Island, NY, USA) and 1 mM EDTA (Sigma) for 30 min at 20–25 °C. After washing, the lung pieces were incubated with 1 mg/mL collagenase (type IV; Sigma), with shaking. The lung mixture was then filtered through a 70-µm pore size nylon cell strainer (BD Falcon, Bedford, MA, USA) and centrifuged for 20 min at 2000 rpm. The cell pellet was collected, and the cells washed twice. BALF was collected as described in the previous section and thereafter, cells were incubated with monoclonal antibodies (mAbs) against CD11b (M1/70, rat DA/HA IgG2b k), and Gr-1 (RB6-8C5, rat IgG2b). All fluorochrome-labelled mAbs and isotype control IgGs were purchased from BD Biosciences (San Diego, CA, USA). Cells from the lungs and BAL cells were incubated with FITC- and PE-labelled mAbs for 30 min, washed with PBS, fixed with 4% paraformaldehyde (Sigma-Aldrich, Korea) for 20 min, washed with PBS, and then stored at 4 °C until analysis by two-color flow cytometry on a FACs caliber (BD Biosciences, Mountain View, CA, USA). By standard methods recommended by Becton Dickinson, two color flow cytometry analysis was performed with scatter gates set on the lymphocyte fraction by forward and side scatter (SCC) and PE or FITC fluorescence FL2 with laser excitation at 488 nm. Dead cells were excluded from analysis by appropriate gating strategies and propidium iodide (PI) staining. The number of immunofluorescence-positive cells (double positive cells) was determined out of 5000 cells analyzed. A cell gate containing lymphocytes was established on the basis of forward and side light scatter. For determination of the borderline between stained and unstained cells, cells were also stained with mouse FITC-conjugated CD11b and PE-conjugated Gr-1. Percentages were calculated on the basis of the number of lymphocytes found in each quadrant. The general status and stability of the cytometer was checked daily. Multi-color analysis and/or quantitation was performed, then the FITC/PE Compensation Standard™ was performed on the run prior to collection of data on the Gr-1 and CD11b population the same day.

### 2.6. BALF and Cytokine Measurements

Mice were anesthetized by an i.p. injection of urethane (100 µL; Sigma-Aldrich, Korea), and their lungs gently lavaged with 1 mL of 0.9% saline via a tracheal cannula. Total and differential BALF cell counts were determined. Samples were centrifuged at 2000 rpm for 10 min, and the supernatants stored at −80 °C. Symmetric dimethyl-arginine (SDMA) in serum, MIP-2 (CXCL-2), IL-17, CXCL-1, and TNF-a in BALF were measured by ELISA using a monoclonal antibody-based mouse interleukin ELISA kit (R&D systems, USA), according to the manufacturer’s instructions.

### 2.7. RNA Extraction and Quantitative Real-Time Polymerase Chain Reaction (qRT-PCR) for mRNA Expression

qRT-PCR was performed after quantitative normalization for each gene by a densitometry using GAPDH gene expression. Total cellular RNA was extracted from the lung by the phenol-chloroform-based method (RNAsolB: Tel-Test Co. Inc., Friendswood, Texas, USA) according to the manufacturer’s instructions. qRT-PCR was performed using the Applied Biosystems 7500 Fast Real-Time PCR system (Applied Biosystems, Foster, CA, USA). Gene expression was analyzed with SYBR Green PCR Mastermix (ABI) and the final primer concentration of 200 nM using GAPDH as the internal standard. The following PCR regimen was used: 2 min at 50 °C, 10 min at 94 °C, and 40 cycles of 1 min at 94 °C, and 1 min at 60 °C. The amount of SYBR Green was measured at the end of each cycle. The cycle number at which the emission intensity of the sample rose above the baseline was referred to as the relative quantification (RQ) and was proportional to the target concentration. qRT-PCR analyses were performed in triplicate and analyzed by applied Biosystems 7500 Fast Real-Time PCR system manual (threshold: 0.05; baseline: 6–15 cycles). Relative quantitative (RQ) evaluations by qRT-PCR were expressed for various samples. PCR primer sequences are listed in Table 1.

### 2.8. Histological Examination

Lungs were infused with 1 mL of 10% neutral formalin and immersed in the same fixative for at least 24 h. Tissues were embedded in paraffin and 6 µm sections were cut and stained with H&E, Mason trichrome, periodic acid–Schiff (PAS), and alcian blue PAS (AB-PAS) staining (obtained from Sigma-Aldrich) to assess cell infiltration and fiber-formation, respectively, under a light microscope. To determine the severity of inflammatory cell infiltration, peri-bronchial cell counts, extent of mucus production, and goblet-cell hyperplasia in the airway epithelium were blindly quantified using the 3-point (0–2) grading system.

### 2.9. Immunohistofluorescent (IHF) Staining

The lung tissues were frozen at −20 °C, and sections cut to a thickness of 20 µm using a Cryostat Microtome (CM 3050S, Leica Microsystems, Wetzlar, Germany). Lung sections (20 µm) were fixed with 4% paraformaldehyde and 4% sucrose in phosphate-buffered saline (PBS) at 20–25 °C for 40 min, permeabilized with 0.5% Nonidet P-40 in PBS, and blocked with 2.5% horse serum and 2.5% bovine serum albumin for 16 h at 20–25 °C. Double immunofluorescence staining was performed by incubating tissue sections with antibodies for STAT3 (Cell Signaling Technology, Inc. USA) overnight at 4 °C. Subsequently, fluorescein-conjugated secondary antibody was added for 2 h, and nuclear staining performed using Hoechst. Sections were observed using an Eclipse Ti-E inverted fluorescent microscope (Nikon Instruments Inc., Mississauga, Canada).

### 2.10. RNA Preparation and qRT-PCR in EL-4 and HMC-1 Cells

GA was dissolved in water as a 10 mM/L stock solution. TN was dissolved in DMSO as a 100 mM/L stock solution. GG and AR, in the ratio 1:4, were dissolved in 1:2 mixed solution of DMSO and PBS. All stock solutions were stored at −20 °C. The stocks were diluted with the same solution of the stocks for experiments. LA4 cells were cultured in RPMI 1640 medium (Gibco BRL, Grand Island, NY) supplemented with 10% fetal bovine serum (FBS), 100 U/mL of penicillin, and 100 µg/mL streptomycin at 37 °C in 5% CO2. LA4 cells were pre-treated with vehicle, GG plus AR (120 or 240 μg/mL), GA, TN for 30 min prior to the addition of 80 µg/mL of CFA for 6 h. HMC-1 cells were cultured in IMDM supplemented with 10% FBS and 1% Gibco^®^ antibiotic antimycotic (containing 100 units/mL penicillin, 100 μg/mL streptomycin, and 0.25 μg/mL amphotericin B) in an incubator at 37 °C and 5% CO2. HMC-1 cells were pre-treated with the vehicle, GG plus AR (100 or 200 μg/mL), GA, and TN for 30 min prior to the addition of 40 nM of PMA plus 1 μM of A23187 (PMACI) for 6 h. The cells were seeded 16 h before RNA preparation.

LA4 and HMC-1 cells were collected 6 h after different drug treatments, and the total RNA(2 μg) was isolated using Ambion TRIzol^®^ reagent purchased from Life Technologies (Gibco-BRL, Grand Island, NY, USA). The total RNA was reverse-transcribed into cDNA using ReverTraAce cDNA Synthesis kit (Toyobo, Osaka, Japan). Quantitative real-time polymerase chain reaction (qPCR) was performed using the StepOnePlus Real-Time PCR system (Applied Biosystems, Foster City, CA, USA). Then, 100 ng of cDNA and 10 pmole of gene-specific forward and reverse primers were loaded into the qPCR system. IL-17A, CXCL2, and GAPDH were amplified using an SYBR Green PCR MasterMix (Applied Biosystems). The primer sequences used were as follows: IL-17A, sense 5′-TCCCTCTGTGATCTGGGAAG-3′, antisense 5′-CTCGACCCTGAAAGTGAAGG-3′; CXCL2, sense 5ʹ-ATCCAGAGCTTGAGTGTGACG-3ʹ, antisense 5ʹ-CCATACCATGCTGCTGTTGCAC-3ʹ; GAPDH, sense 5ʹ-AACGGATTTGGCCGTATTGG-3′, antisense 5ʹ-GCCTTGACTGTG CCGTTGAA-3ʹ. The oligonucleotides were manufactured by Genotech (Geno Tech Corp., Daejon, Korea). The TaqMan probes for CXCL2 (Hs00236966_mL) were selected using Assays-On-Demand Gene Expression Products (ABI).

The following amplification protocol was used: 2 min at 50 °C, 2 min at 95 °C, then 40 cycles of 15 sec at 95 °C, and 1 min at 62 °C. The target-genes expression was determined (threshold cycle, Ct) and normalized to the Ct of GAPDH. For each gene, qPCR was performed in triplicate and quality controlled by verifying a single peak in the melting-curve analysis; the comparative-threshold (C_T_) method (ΔΔC_T_ method) was used to examine the relative quantification of the samples (Relative Quantitation computer software, Applied Biosystems). Fold-expression changes were calculated using the equation 2^−ΔΔCT^.

### 2.11. Identification of GA, TN, GG, AR, and GG Plus AR by High-Performance Liquid Chromatography (HPLC) Analysis

The samples were analyzed by high-performance liquid chromatography (HPLC) using Agilent infinity 1260 II (Agilent, Santa Clara, Ca, USA) equipped with an autosampler, a quaternary pump, column temperature control, and DAD detector. Tilianin and glycyrrhizic acid were analyzed using a Phenomenex (USA) Luna^®^ C18 (5 μm column, 250 mm × 4.6 mm) column; a suitable guard column (C18, 5 μm, 7.5 mm × 4.6 mm) was used for all chromatographic separations. The mobile phase was composed of 10% acetonitrile with 1.5% (v/v) HAc and 20 mM NH4Ac for solution A and 90% acetonitrile with 1.5% (*v*/*v*) HAc and 20 mM NH4Ac for solution B. The elution profile was: 0–20 min, 25–38% B in A (linear gradient); 20–23 min, 38–100% B in A (linear gradient); 23–25 min, 100–25% B in A (linear gradient), 25–30 min, 25% B in A (isocratic manner). The separation temperature was maintained at 40 °C throughout the analysis, with a flow rate of 1 mL/min. Peaks were detected at 250 nm. The injection volume was 10 μL. Sample peaks were assigned according to retention time and the UV spectra of the four standard compounds in the chromatogram.

For the calibration curves, the working solution was prepared by diluting the stock solution as appropriate. The stock solutions were prepared by dissolving standard compounds in methanol to obtain four or five concentrations of working solutions. Particularly for tilianin, the standard compound was dissolved in a small amount of DMSO and sonicated to aid dissolution before adding methanol. The calibration curves of tilianin and glycyrrhizic acid were constructed by plotting the peak areas against the concentrations of respective working solutions using linear regression analysis.

### 2.12. Statistical Analysis

Data were analyzed by one-way analysis of variance (ANOVA) and Duncan’s multiple comparison test were applied using Prism v. 7.0 (GraphPad Inc., San Diego, CA). The results are presented as mean *±* SEM, and significant differences denoted as: *# P*< 0.05, *## P*< 0.01, and *### P*< 0.001 (compared to normal), and * *P*< 0.05, ** *P* < 0.01, and *** *P* < 0.001 (compared to control).

## 3. Results

### 3.1. Chemical Analysis of CFA

The results of the analysis of representative toxic chemicals generated during coal combustion at 700 °C are listed in Figure 1A. Benzo(α)pyrene was present in the highest amount of 83.39 µg/mg. Among phenol compounds, the content of meta/para (m + p)-cresol, ortho(o)-cresol, phenol, catechol, hydroquinone, and resorcinol was high, in the order given above. The content of 1-naphthylamine, 2-naphthylamine, 3-aminobiphenyl, and 4-aminobiphenyl aromatic amines was 1.36, 1.72, 0.21, 0.16 (µg/mg), respectively. The major components of CFA are silica, ferrous oxide, aluminum oxide, calcium oxide, magnesium oxide, and carbon in varying amounts as measured by a loss on ignition (LOI) test [18]. In general, it can be seen from the summary data in Figure 1B that CFA contains a variety of metal oxides in the order SiO2 > Al2O3 > Fe2O3 > MgO > K2O > TiO2 > CaO. Carbon is one of the most plentiful chemicals in airborne particles. Particulate matter (PM) or fine particulate matter containing carbon has become a significant issue influencing human health and climate change. Elemental carbon (EC) is a primary pollutant, implying that it is directly emitted into the atmosphere from anthropogenic or natural sources, while the presence of organic carbons (OC) can be a primary pollutant (when in particulate form) or secondary (when present as a gas). When the OC and EC amounts were expressed in μg/m^3^, CFA contained 1592 μg/m^3^ of OC and 102 μg/m^3^ of EC.

### 3.2. Inhibitory Effect of GG, AR, GG plus AR, GA, and TN on Neutrophil Accumulation in BALF and Lymphocytes Recruitment in the Lungs of Murine COPD Model

COPD progression is associated with the recruitment of T lymphocytes mainly into the small airways. The number of neutrophils, lymphocytes, and macrophages are increased in the lungs of smokers and patients with COPD [19]. Neutrophils are a particularly important factor in the development and pathogenesis of COPD; hence, the inflammation that occurs in COPD is described as neutrophilic. To evaluate the patterns and magnitudes of inflammatory response in the lungs and BALF after exposure to CFD (CFA plus DEP), mice were exposed intratracheally to CFD preparations as described above. Based on anti-inflammatory activities of GG, AR, GG plus AR, GA, and TN, we investigated the effect of these compounds on lung inflammatory responses in the mice COPD model. As shown in Figure 2, CFD exposures increased the total BALF cells and neutrophils in BALF and the lungs and caused inflammation in the mice airway and the lungs (Figure 3). Representative microscopic images of BALF content are shown in Figure 2E. Administration of middle or high concentration of GG plus AR (100 and 200 mg/kg) significantly reduced CFD-mediated inflammatory cell recruitment, whereas low concentration of GG plus AR (50 mg/kg) had only a slight effect on cell recruitment. Moreover, administration of GA and TN, the major active components of GG and AR, decreased the total BALF cells and neutrophils in BALF.

### 3.3. Effects of GG, AR, GG Plus AR, GA, and TN on Histopathology of Lung Injury in Murine COPD Model

As shown in Figure 3A, typical histopathological features of COPD, namely distinct recruitment of leukocytes, goblet cell hyperplasia, erosion in perivascular and peribronchial areas, damaged alveolar walls, and pulmonary bullae were found in the lungs of control-group mice exposed to CFD. Migrating inflammatory cells such as neutrophils and eosinophils, were observed mainly in the peribronchial regions of the lungs. Histological sections from the dexamethasone-, GG-, AR-, GG plus AR- (middle and high concentration), and TN-treated mice showed reduced airway inflammation in lung tissues (Figure 3A). The airway inflammation was not decreased effectively in histological sections of the lung tissue from GG plus AR (low concentration)- and GA-treated mice.

Compared to the control group, exposure to CFD resulted in a significant increase in subepithelial fibrosis, collagen deposition in perivascular, and peribronchial tissue (Figure 3B), which were reduced by GG, AR, GG plus AR (middle and high concentration), and TN administration (Figure 3B). This result is consistent with the H&E staining results described above. Goblet-cells hyperplasia from airway epithelium of CFD-exposed mice were attenuated by GG, AR, GG plus AR (middle and high concentration), and TN treatment (PAS positive staining area) as identified by PAS staining (Figure 3C), AB-PAS staining (Figure 3D). These results suggest that GG, AR, GG plus AR (middle and high concentration), and TN administration ameliorates CFD-induced histopathological changes in the mice lungs. The overall effect of GG plus AR combination was greater than the combined individual effects of separate GG and AR.

### 3.4. Inhibitory Effects of GG, AR, GG plus AR, GA, and TN on Inflammatory Chemokines, Cytokines in BALF, and SDMA in Serum of Murine COPD Model

Based on anti-inflammatory activities of GG, AR, GG plus AR, GA, and TN, we evaluated the effect of these compounds on the production and levels of inflammatory cytokines in the lungs of mice in the murine COPD model. As shown in Figure 4, exposure to CFD caused inflammation in the mouse airways and the lungs, manifested by higher levels of CXCL-2 (Figure 4A), IL-17A (Figure 4B), CXCL-1 (Figure 4C), TNF-α (Figure 4D), and SDMA (Figure 4E).

Symmetric-dimethylarginine (SDMA) levels were reported to correlate with the airway neutrophilic inflammation, suggesting an association between airway inflammation in COPD and endothelial dysfunction [20]. Increased expression of CXCL1, CXCL2, and their receptor CXCR2 after exposure to air pollutants such as CS, DEP, and CFA suggests that the pollutants play a role at least in part in attracting inflammatory cells. The above receptors have been found to recognize neutrophil chemokines CXCL1(KC or GROα) and CXCL2(GROβ or MIP-2) [21].

In our study, animals exposed to CFD showed increased levels of CXCL-2, IL-17A, CXCL-1, and TNF-α in BALF and SDMA in serum as compared to those seen in the untreated normal group. Administration of AR, GG plus AR (middle and high concentration), GA, and TN significantly suppressed the increased levels of these chemokines as compared to the untreated CFD-induced COPD-model mice (Figure 4). Administration of AR attenuated CXCL-2, IL-17A, and TNF-α levels in BALF, however, no inhibition of CXCL1 and SDMA production after AR treatment was seen. In addition, GA administration (10 mg/kg) had no effect on SDMA production. Individual herb extracts (especially AR) and active components (GA, TN) showed an attenuating effect on the production of inflammatory cytokines and chemokines; interestingly, treatment with herbal mixture (GG plus AR) exerted less effect than the individual herbal treatments (Figure 4).

### 3.5. Inhibitory Effects of GG, AR, GG plus AR, GA, and TN on mRNA Expression Of Inflammatory Cytokines and TRPV1 in the Lung Tissue of the Murine COPD Model

Many inflammatory mediators can trigger production of MUC5AC. IL-17 plays a role in upregulating mucus production in the airway. MUC5AC, a major mucin protein that constitute much of the mucus in respiratory secretions, is inducible by IL-17 [22]. TNF-α and IL-6 are well known to be pro-inflammatory cytokines that contribute to pollutants-induced lung inflammation. The antioxidant effects of our potential materials were confirmed by a decrease of inflammatory cytokines genes IL-6, TNF-α, nitric oxide synthase gene (NOS-II), IL-1β (data not shown). Potentially transient receptor vanilloid-1 (TRPV1) is also expressed by epithelial cells of the airways and alveoli, and activation of TRPV1 in these cells by pollutants has been correlated with the production of immunomodulatory cytokines and chemokines including IL-6, IL-8, and TNF-α [23]. CFA-induced expression of CXCL-1, CXCL-2, and IL-6 were reduced in TRPV1 knock-out mice [24]. Therefore, we assessed the CXCL2, IL-17A, CXCL1, MUC5AC, IL-6, TNF-α, NOS-II, and TRPV1 mRNA gene expressions in the lungs using quantitative real-time PCR. The CFD-induced control mice showed increased expression of CXCL2, IL-17A, CXCL1, MUC5AC, IL-6, TNF-α, NOS-II, and TRPV1 in lung tissues compared with the normal group. Notably, administration of GG plus AR (middle and high concentration) significantly suppressed CXCL2, IL-17A, CXCL1, MUC5AC, IL-6, TNF-α, NOS-II, and TRPV1 gene expression compared to the control group (Figure 5). The gene expression of CXCL1 in CFD-induced control groups was significantly higher than the normal group (Figure 5C). Except for the NOS-II expression, GA (10 mg/kg) had no effect on CXCL2, IL-17A, CXCL1, MUC5AC, IL-6, TNF-α, and TRPV1 gene expression in these experiments. On the other hand, administration of TN (10 mg/kg) significantly reduced the gene expressions of CXCL2, IL-17A, MUC5AC, IL-6, TNF-α, NOS-II, and TRPV1 compared to the control but not of CXCL1. Consistent with the levels of cytokines in BALF reported above, not all target genes were affected but the effect of a combination (GG plus AR, middle and high dosage) was greater than the sum effect of the individual agents.

### 3.6. Inhibitory Effect of GG, AR, GG plus AR, GA, and TN on CD11b+Gr-1+(high) Neutrophils Infiltration in the Lung Tissue and BALF of the Murine COPD Model

The pathological features of COPD include persistent lung inflammation with infiltration of inflammatory cells including neutrophils. Flow cytometric analysis was used to investigate the effect of GG, AR, GG plus AR, GA, and TN on CD11b+Gr-1+(high) neutrophils in the COPD model. The proportion of CD11b+Gr-1+(high) neutrophils in the BALF and the lung cell of CFD-induced COPD in mice was higher than that in the normal control group, whereas the proportion of these cells in mice treated with GG, AR, GG plus AR, GA, and TN was lower than that in the CFD-induced control mice (Figure 6A–D). However, no significant differences were observed in any of the CD11b+Gr-1+(high) neutrophils population in the BALF of the control and the GA-treated mice (Figure 6B). The expansion of neutrophils in the BALF and the lung cells of the CFD-induced control mice correlated with the increase of CXCL2, IL-17A, and CXCL1 (neutrophil chemokines and cytokine) in the BALF (Figure 4 and Figure 5). Consistent with the other results reported above, the combination of GG and AR (especially the middle-range dosage) are more effective than either component acting alone.

### 3.7. Inhibitory Effect of GG, AR, GG plus AR, GA, and TN on STAT3 Transcription Factor in The Lung Tissue of Murine COPD Model

IL-17 plays a crucial role in the neutrophil infiltration in the lungs. In addition, IL-17 has been shown to be associated with the development of COPD [25]. IL-17 plays a key role in upregulating mucus production such as MUC5AC in the airways, and IL-17 promotes MUC5AC expression and goblet cell hyperplasia [26]. It is well known that STAT3 is the essential transcription factor for Th17 differentiation. Depletion of STAT3 specifically in T cells resulted in a failure of Th17 polarization and reduced IL-17 production [27]. Due to the inhibitory effects of the GG, AR, GG plus AR, GA, and TN on the production of IL-17 cytokine and other chemokines, we applied immunofluorescence analysis to examine the STAT3 signaling pathway involvement in the above processes. As shown in Figure 7, exposure to CFD resulted in the up-regulation of the STAT3 signaling-related protein expression in mice, suggesting that CFD might induce COPD through the STAT3 signaling pathway. By contrast, GG, AR, GG plus AR (middle and high dosage), and TN lowered the level of expression of the STAT3 pathway. Treatment with GG, AR, GG plus AR (middle and high dosage), and TN exerted a protective effect against COPD possibly via the STAT3/IL-17 pathway. However, we did not detect any inhibition of STAT3-protein level by GG plus AR (low dosage) treatment. Moreover, GA (10 mg/kg) had no effect on STAT3-protein expression. Interestingly, using combinations of GG and AR (especially middle or high dosage) is more effective than applying AR alone, but not GG.

### 3.8. Inhibitory Effect of GG Plus AR, GA, and TN on CXCL2 and IL-17A mRNA Gene Expression In Vitro (LA4 and HMC-1 Cell Line)

To confirm the underlying mechanisms of CXCL2 and IL-17A gene expression and neutrophil migration in vivo, we tested the mRNA gene-expression levels of CXCL2 and IL-17A in the mouse epithelial cell line LA-4 and human mast-cell line HMC-1 using qRT-PCR.

The expression levels of CXCL2 and IL-17A were significantly up-regulated in CFA-treated control cells compared to normal cells (Figure 8A,B). Consistent with the mRNA and protein levels of CXCL2, IL-17A in the murine COPD model (Figure 4 and Figure 5), treatment with GG plus AR, GA, and TN in LA4 cells and HMC-1 cells inhibited the expression of CXCL2 and IL-17A, the cytokines that recruit neutrophils to inflammatory tissues, without causing any toxicity (as determined by MTT assay, data not shown) (Figure 8A,B), whereas CXCL2 and IL-17A levels in LA-4 cells were not decreased by the treatment with GA (low dosage) compared with the CFA control. There was also no significant decrease of IL-17A in the HMC-1 cell line (data not shown). The mechanisms involved in these experiments are not clear, so further work will be needed to investigate the inhibitory effect of GG plus AR, GA, and TN on CXCL2 and IL-17A mRNA levels in different cell types. Collectively, these results confirmed that GG plus AR, GA, and TN suppressed neutrophil infiltration by affecting the CXCL2 and IL-17A gene expression.

### 3.9. Identification of the Main Components in GG, AR, and Their Mixture by HPLC Analysis

HPLC fingerprint was generated to identify the main chemical constituents in GG, AR, and their mixture. Two major chemical constituents were identified from HPLC chromatogram based on UV spectrograms and retention times of the corresponding commercial reference standards. (Figure 9).

Two major components were determined from major peaks by comparison of retention time and UV spectra on HPLC chromatogram with commercial standards. The fingerprint of GG, AR, and their mixture showed a chromatographic profile with peaks with retention times of 9.67 and 17.28 min that were ascribed to the presence of GA and TN (Figure 9).

## 4. Discussion

COPD is a chronic irreversible and progressive inflammatory disease associated with the airway remodeling, emphysema, airflow obstruction, small-airway thickening, metaplasia of the epithelium, subepithelial fibrosis, increased goblet-cell size, and mucus hypersecretion [28] mediated by the increased levels of cytokines, chemokines, and inflammation-associated cell receptors [29].

COPD etiology is usually linked to cigarette smoking or environmental-pollutant exposure such as coal fly ash and diesel-exhaust particles. Several researchers examined complex chemical mixtures such as CFA and DEP as materials that may represent the complex make-up of urban-collected PM [30,31]. Several investigators reported a significant lung-inflammation response to airway exposure to CFA and DEP [32,33,34]. Numerous different types of airborne particles including cigarette smoke (CS), carbon black (CB), CFA, and DEP are typical constituents of urban PM and are commonly used in studies to investigate effects of PM in airway-disease models. Previous reports showed that various types of PM can induce variable levels of airway inflammation and are an important inducer of BAL-neutrophil accumulation. Importantly, urban PM as well as surrogate PM particles such as CFA and DEP showed a similar effect on the induction of neutrophils [35]. The role of neutrophils, macrophages, and other cells and their products has been well established in the progression of COPD. In response to IL-17A and IL-17RA signaling, cells produce several proinflammatory chemokines and cytokines. Among the chemokines that are upregulated are CXCL1 and CXCL2 which are chemoattractants for neutrophils [36]. GG, one of the most important and popular herbal medicines in the world, has been widely used in traditional Korean medicine as a cough reliever and as an immunomodulatory, anti-inflammatory, and detoxifying agent. So far, numerous diverse compounds have been isolated from GG such as triterpene saponins and flavonoids. Glycyrrhizic acid (GA), a triterpenoid saponin, is the most important constituent of GG. Previously described, GA has a wide range of pharmacological effects including anti-inflammatory, antiallergy, as well as the immune-system-activating properties [37,38,39]. Similarly, AR has been used as a traditional medicine to treat various diseases based on its various pharmacological activities which include anti-inflammatory, antioxidant, and anti-atherogenic effects [40,41]. AR contains flavonoids including tilianin, rosmarinic acid, and acacetin which contribute to antioxidant and anti-inflammatory effects [9,10,42]. In our study, as part of an ongoing search for potential anti-COPD agents, TN was used as a representative standardized compound of AR. In particular, it was previously reported that tilianin inhibited MUC5AC expression mediated via modulating the EGFR-MEK-ERK-Sp1 signaling pathway in human-airway epithelial cells [43], and effectively regulated inflammatory cytokines such as TNF-α, IL-1β, IL-18, and MCP-1 [44]. However, the mechanisms by which GG, AR, and their mixture inhibit neutrophilic lung inflammation has not been fully elucidated; in our study, we employed a murine COPD model for such purpose. Traditionally, it is well known that herbal combinations are more effective than the individual herbs used alone and it has been usually believed that synergistic interactions between the components of herb mixtures are a crucial part of their therapeutic efficacy. In general, synergistic effects are thought to be positive, and together with low concentrations used, are recognized as benefits. Synergic effects occur if two or more individual herbal medicines mutually enhance each other’s effect more significantly than the simple sum of these individual-herbs effects [45]. In our study, we used the same approach although it is apparent that there may also be negative aspects.

As shown in Figure 2 and Figure 3, CFD exposure induced an inflammatory process in the BALF and the lungs parenchyma. Moreover, CFD exposure increased the epithelial thickness and resulted in a distinct recruitment of leukocytes, increased goblet cell hyperplasia, erosion in perivascular and peribronchial areas, damaged alveolar wall and pulmonary bullae. Collagen deposition in perivascular and peribronchial tissue (Figure 3B) was reduced by GG, AR, GG plus AR (middle and high concentration), and TN administration (Figure 3B). This result is consistent with the H&E stain results reported above. Goblet-cells hyperplasia of the airway epithelium of CFD-exposed mice was attenuated by GG, AR, GG plus AR (middle and high concentration), and TN treatment (PAS positive staining area) and was evidenced by PAS staining (Figure 3C). The effect of combined GG plus AR was greater than that of the individual (GG or AR) components.

As previously mentioned in the results, SDMA levels correlate with airway neutrophilic inflammation, suggesting an association with airway inflammation in COPD. CXCL1, CXCL2, and IL-6 are important chemo/cytokines for neutrophils. Increased expression of CXCL1, CXCL2, and of their receptor CXCR2 elicited by exposure to air pollutants play a pivotal role at least in part in attracting inflammatory cells. The CXCR2 attracts and bind neutrophil chemokines CXCL1 and CXCL2. These reports are consistent with the higher expression of CXCL1 and CXCL2 observed in lung tissues in our study. In the CFD control group, the levels of CXCL-2, IL-17A, CXCL-1, TNF-α in BALF, and SDMA in serum were higher than those of the untreated normal group. Administration of AR, GG plus AR (middle and high concentration), GA, and TN significantly suppressed the increased levels of these components compared to the levels seen in the CFD-induced murine COPD model (Figure 4).

Administration of AR attenuated CXCL-2, IL-17A, and TNF-α production in BALF, although it did not inhibit CXCL1 and SDMA production after AR treatment. Moreover, GA (10 mg/kg) had no effect on SDMA production. IL-17 plays a crucial role in the neutrophil response in the lungs. Recombinant IL-17A administration to the lungs induces a powerful neutrophil recruitment [29].

Our results demonstrate that the IL-17A/IL-17 RA axis is important to murine airway fibrosis and inflammation. Our findings suggest that IL-17 might be targeted to prevent the progression of airway fibrosis in COPD. GG, AR, and their mixture could be a potential anti-inflammatory drug for COPD. In particular, their mixture is a valuable candidate worthy of further evaluation for future diverse PM-induced COPD therapy. MUC5AC is an important gene expressed by airway epithelial goblet cells [46]. Stimuli such as cigarette smoke [47] and chemokines [48] upregulate MUC5AC production and induce goblet-cells metaplasia or hyperplasia. IL-17 plays a role in upregulating mucus production in the airways. MUC5AC, a major mucin protein that constitutes much of the mucus in respiratory secretions, is inducible by IL-17 [22]. IL-6, TNF-α, and IL-1β also play an important role in the pathogenesis of emphysematous lung [49]. Therefore, inhibition of these proinflammatory cytokines is one of the most prospective treatments for COPD [50].

The antioxidant effects of our various potential materials were confirmed by decrease expression of inflammatory-cytokines genes IL-6, TNF-α, nitric oxide synthase gene (NOS-II), and IL-1β (data not shown). TRPV1 plays a crucial role in the cough reflex. TRPV1 SNPs are significantly related to cough [51]. TRPV1 agonists dose-dependently inhibit neutrophil counts in the BALF of the LPS-induced acute lung injury model [52]. It is likely that activating TRPV1 might attenuate lung inflammation in patients with COPD. TRPV1 is also expressed by epithelial cells of the airways and alveoli, and activation of TRPV1 in these cells by pollutants has been correlated with the production of immunomodulatory cytokines and chemokines including IL-6, IL-8, and TNF-α. CFA-induced expression of CXCL-1, CXCL-2, and IL-6 was reduced in TRPV1 knock-out mice. Enhanced sensitivity to capsaicin and excessive expression of TRPV1 were observed in airway diseases accompanied by cough [53]. Therefore, we assessed the CXCL2, IL-17A, CXCL1, MUC5AC, IL-6, TNF-α, NOS-II, and TRPV1 mRNA gene expressions in the lungs using quantitative real-time PCR.

Notably, administration of GG plus AR (middle and high concentration) significantly suppressed CXCL2, IL-17A, CXCL1, MUC5AC, IL-6, TNF-α, NOS-II, and TRPV1 gene expression compared to the control group (Figure 5). Administration of TN (10 mg/kg) significantly reduced the gene expressions of CXCL2, IL-17A, MUC5AC, IL-6, TNF-α, NOS-II, and TRPV1 compared to the control group; however, it did not attenuate CXCL1 gene expression. Consistent with the above-reported levels of cytokines in BALF, for most but not all target genes, the effect of combinations (GG plus AR, middle and high dosage) is greater than that of each of the individual components.

The proportion of CD11b+Gr-1+(high) neutrophils in the BALF and lung cells of CFD-induced COPD murine model was higher than that in the untreated group, whereas the proportion of CD11b+Gr-1+(high) neutrophils in GG, AR, GG plus AR, GA, and TN-treated mice was lower than that in the CFD-induced control mice (Figure 6A–D). The expansion of the neutrophils in the BALF and lung cell of CFD-induced control mice correlated with the increase of CXCL2, IL-17A, and CXCL1 (neutrophil chemo/cytokines) in the BALF (Figure 4 and Figure 5). Consistent with other results reported above, combinations of GG and AR (especially middle dosage) are more effective than either of the components alone. Th17 cells release IL-17A which acts on airway epithelial cells to release CXCL1 and CXCL2 that in turn attracts neutrophils. Th17 cells also release IL-21 which promotes Th17-cell differentiation via the STAT3 transcription factor [29]. Activation of STAT3 also induces the expression of another related nuclear receptor, RORα, which synergizes with RORγt in Th17 cell differentiation [54]. We acknowledge that the present study did not evaluate the RORγt expression in the lung tissue of the murine COPD model. Although we showed that AR and its mixtures decreased the number of neutrophils and simultaneously attenuated IL-17 expression in the BALF and the lungs, we did not evaluate the direct relationship between neutrophilic inflammation and RORγt/IL-17. However, previous studies reported an indirect relationship suggesting that GA could reduce the expression of RORγt and IL-17A [55]. Glycyrrhizae radix reduce Aβ secretion by increasing PPARγ expression and inhibiting STAT3 phosphorylation [56].

NF-kappaB and STAT-3 activation was significantly reduced by glycyrrhizic acid treatment [57]. Total flavonoids from glycyrrhizae radix exert anti-inflammatory effects by inactivating iNOS via JAK2/STAT3 signaling pathways [58]. Thymoquinone, a main aromatic component of AR, inhibited STAT3 phosphorylation on APAP-induced acute liver injury [59].

Consistent with our findings, a recent study reported that STAT3 is activated in the lungs obtained from patients suffering from severe COPD. Activation of the STAT3 pathway is critical for persistent inflammation in lung tissues [60]. Ablation of STAT3 specifically in T cells resulted in a failure of TH17 polarization and reduced IL-17 production. An increasing body of evidence shows the requirement of STAT3 for Th17 cell development. Indeed, in mice, STAT3 is absolutely required for the induction of IL-17, IL-17F, and RORγt [61]. IL-17 is produced by inflammatory cells and targets structural cells such as epithelial cells [62]. Mast cells release neutrophil chemoattractants CXCL1 and CXCL2 in both granular and newly synthesized forms [63]. To confirm the underlying mechanisms of CXCL2 and IL-17A gene expression and neutrophil migration in vivo (Figure 4A,B, Figure 5A,B), we tested the mRNA gene-expression levels of CXCL2 and IL-17A in the mouse epithelial cell line LA-4 and human mast-cell lines which are the main sources of these cytokines (Figure 8A–E). As seen in Figure 7, CFD caused the up-regulation of the STAT3 signaling-related protein expressions in mice, suggesting CFD might induce COPD through the STAT3 signal pathway. By contrast, GG, AR, GG plus AR (middle and high dosage), and TN inhibited the expression levels of the STAT3 pathway. Herein, GG, AR, GG plus AR (middle and high dosage), and TN exerted the protective effect against COPD possibly via the STAT3/IL-17 pathway. However, GG plus AR at low dosage did not inhibit STAT3 protein expression. Moreover, GA (10 mg/kg) had no effect on STAT3-protein expression. Further, combinations of GG and AR (especially middle or high dosage) was more effective than AR alone, but not GG. The ability of GG, AR, and their derivatives including glycyrrhizic acid and tilianin have been examined to determine their possible effect on immunomodulatory and anti-inflammatory conditions such as those seen in COPD and asthma. Our study provided evidence that administration of the individual herbs, their ingredients, and their mixtures reduced physiological and pathological manifestations in the lungs of mice exposed to CFD. This effect was associated with the reduction of CFD-induced CXCL2, IL-17 production, and inflammatory responses in vivo in mice and in vitro in cells (Figure 8A–E). In our study, CFD caused the up-regulation of the STAT3/IL-17/CXCL2 signaling-related protein expressions in mice and cells, suggesting CFD might induce COPD through STAT3/IL-17/CXCL2. By contrast, in CFD-exposed mice and cells, AR (100 mg/kg), GG plus AR (100, 200 mg/kg), TN (10 mg/kg), and GA (10 mg/kg) inhibited expression levels of STAT3/IL-17 and CXCL2 pathways. However, how these herbs and herbal components interact with each other, synergistically or antagonistically, both at the molecular and systems levels, remains unclear. Synergistic activities are responsible for the therapeutic efficacy of a considerable number of herbal medicines and their mechanisms may be investigated by analyzing the component-mediated molecular interactions and network actions.

Probably, more herbal ingredients of the mixture of GG and AR mutually enhance each other’s effect more significantly than the simple sum of these components or individual agents including tilianin and glycyrrhizic acid. The potential synergistic mechanisms of the mixture of GG and AR can be explained by enhancing each other’s activity, and interacting with multiple targets of a pathway and its crosstalk pathways. Herbal products contain plant extracts, which are complex mixtures of various compounds. As with traditional drugs, it is necessary to validate their efficacy and safety through preclinical and clinical studies. Collectively, the mixture of GG and AR synergistically exerted the protective effect against COPD possibly via the STAT3/IL-17 and CXCL2 pathways.

## 5. Conclusions

In conclusion, the study results indicate that GG, AR, and their mixtures could exert therapeutic effect on CFD-induced COPD; the effect might be attributed to the inhibition of the CXCL2 and STAT3/IL-17 pathways. Herbal combinational medication of GG and AR (100 mg/kg), in the ratio 1:4 by weight, is beneficial and most effective. Therefore, herbal combinational medication of GG and AR (100 mg/kg) have a potential to be developed into therapeutics for the treatment of COPD by regulating CXCL2/IL-17 production via modulation of the STAT3 pathway. Our work could produce a possible new therapeutic application.

## Figures and Tables

**Figure 1 nutrients-12-00926-f001:**
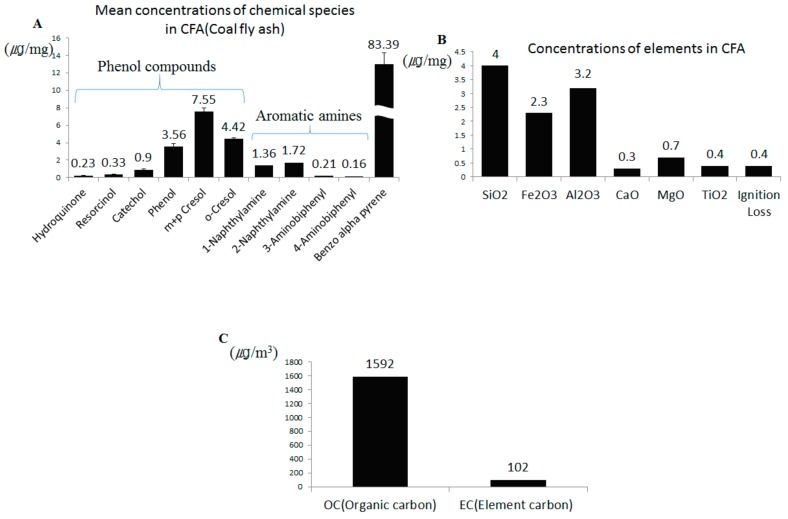
Constituents of polycyclic aromatic hydrocarbons (PAHs) content in coal fly ash (CFA) combustion particles (**A**). Chemical composition and mean concentrations of chemical species of CFA (**B**). Composition of OC (organic carbon), EC (element carbon) in CFA sample (**C**).

**Figure 2 nutrients-12-00926-f002:**
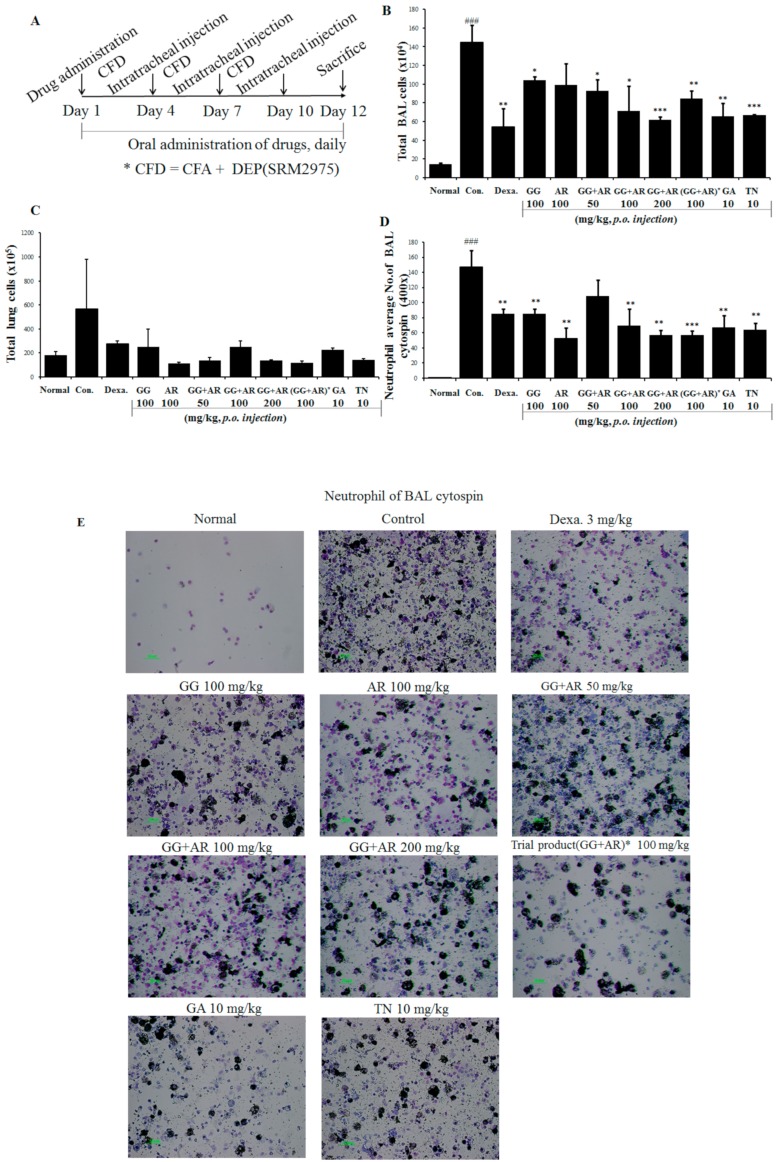
Schematic diagram of CFD-induced chronic obstructive pulmonary disease (COPD) model mice (**A**). Total bronchoalveolar lavage (BAL) cell numbers of each treatment group (**B**). Total lung cell numbers of each treatment group (**C**). Neutrophil numbers in the BALF cytospin of each treatment group (**D**). Neutrophilic inflammation cytospin image (×400 images of cytospin slide) shown by differential leukocyte staining from BAL slides (**E**). Normal: normal group; Con.: CFD, CFD-induced COPD model group; Dexa.: CFD solution and administrated with dexamethasone 3 mg/kg group; GG 100: CFD solution and administrated with GG extract 100 mg/kg group; AR 100: CFD solution and administrated with AR extract 100 mg/kg group; GG + AR 50: CFD solution and administrated with GG plus AR extract 50 mg/kg group; GG+AR 100: CFD solution and administrated with GG plus AR extract 100 mg/kg group; GG+AR 200: CFD solution and administrated with GG plus AR extract 200 mg/kg group; GG+AR 100* (trial product): CFD solution and administrated with GG plus AR (trial product) 100 mg/kg group; GA 10: CFD solution and administrated with GA 10 mg/kg group; TN 10: CFD solution and administrated with TN 10 mg/kg group. The statistical significance of differences between control and treatment groups were assessed by ANOVA and Duncan’s multiple comparison test were applied using Prism v. 7.0 (GraphPad Inc., San Diego, CA). Data represent the mean ± SEM of 8 independent experiments. (### *P* < 0.001, significantly different from the value of normal group.* *P* < 0.05, ** *P* < 0.01 and *** *P* < 0.001 significantly different from the value of control group.).

**Figure 3 nutrients-12-00926-f003:**
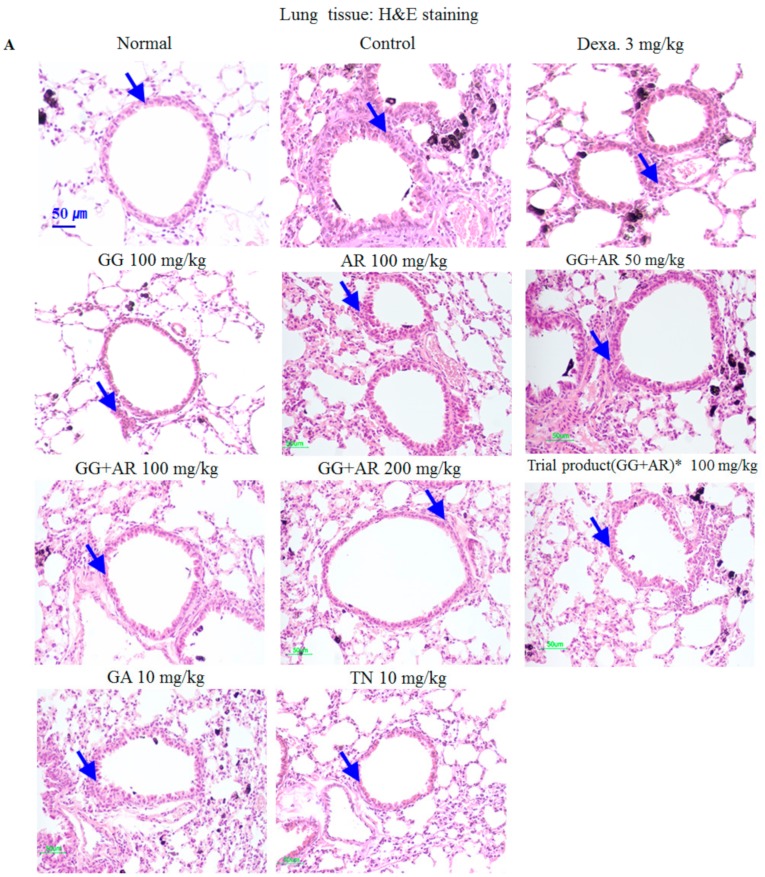
Inhibitory effects of *Glycyrrhiza glabra* (GG), *Agastache rugosa* (AR), GG plus AR, glycyrrhizic acid (GA) and tilianin (TN) on histophathological markers of lung tissue of CFD-induced COPD model mice and drug-treated of each group. Representative H&E staining (**A**), M-T staining (**B**), PAS staining (**C**), AB-PAS staining (**D**), and quantitative analyses of the degree of lung tissue damage (**E**), which denote airway inflammation, subepithelial fibrosis, and goblet cell hyperplasia, respectively. Normal: normal group; Con.: CFD, CFD-induced COPD model group; Dexa.: CFD solution and administrated with dexamethasone 3 mg/kg group; GG 100: CFD solution and administrated with GG extract 100 mg/kg group; AR 100: CFD solution and administrated with AR extract 100 mg/kg group; GG + AR 50: CFD solution and administrated with GG plus AR extract 50 mg/kg group; GG + AR 100: CFD solution and administrated with GG plus AR extract 100 mg/kg group; GG + AR 200: CFD solution and administrated with GG plus AR extract 200 mg/kg group; GG + AR 100* (trial product): CFD solution and administrated with GG plus AR (trial product) 100 mg/kg group; GA 10: CFD solution and administrated with GA 10 mg/kg group; TN 10: CFD solution and administrated with TN 10 mg/kg group. The statistical significance of differences between control and treatment groups were assessed by ANOVA and Duncan’s multiple comparison test. (### *P* < 0.001, significantly different from the value of normal group. ** *P* < 0.01 and *** *P* < 0.001 significantly different from the value of control group.). Blue arrows indicate cell infiltration and inflammation (**A**), fibrosis (**B**), goblet cells (**C**), and mucus hypersecretion (**D**).

**Figure 4 nutrients-12-00926-f004:**
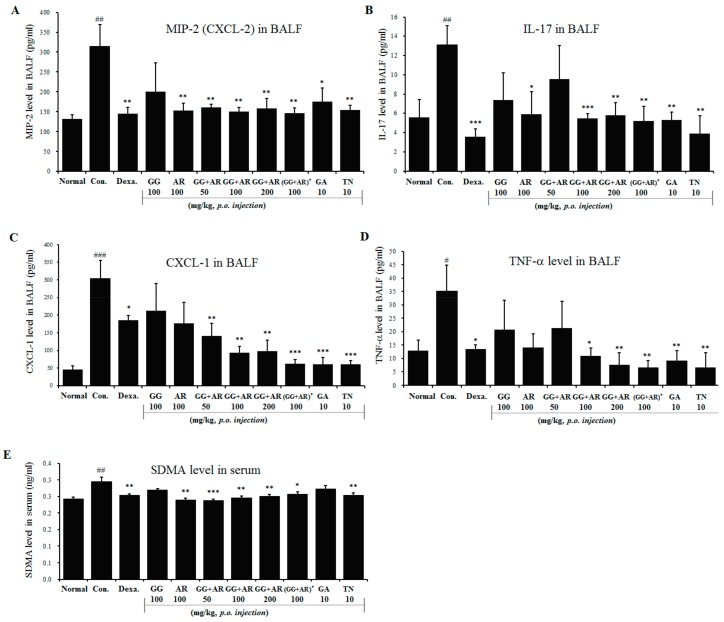
Effect of GG, AR, GG plus AR, GA, and TN on CXCL2, IL-17A, CXCL1, and TNF-α in BALF and SDMA production in serum of COPD model mice. Mice were challenged by intratracheal injection of CFD, and then treated with GG, AR, GG plus AR, GA, and TN over a period of 12 days. The levels of CXCL2 (**A**), IL-17A (**B**), CXCL1 (**C**), and TNF-α (**D**) in BAL fluid and SDMA (**E**) in serum were determined by ELISA. Normal: normal group; Con.: CFD, CFD-induced COPD model group; Dexa.: CFD solution and administrated with dexamethasone 3 mg/kg group; GG 100: CFD solution and administrated with GG extract 100 mg/kg group; AR 100: CFD solution and administrated with AR extract 100 mg/kg group; GG + AR 50: CFD solution and administrated with GG plus AR extract 50 mg/kg group; GG + AR 100: CFD solution and administrated with GG plus AR extract 100 mg/kg group; GG + AR 200: CFD solution and administrated with GG plus AR extract 200 mg/kg group; GG+AR 100*(trial product): CFD solution and administrated with GG plus AR (trial product) 100 mg/kg group; GA 10: CFD solution and administrated with GA 10 mg/kg group; TN 10: CFD solution and administrated with TN 10 mg/kg group. The statistical significance of differences between control and treatment groups were assessed by ANOVA and Duncan’s multiple comparison test. Data represent the mean ± SEM of 8 independent experiments. (# *P* < 0.05, ## *P* < 0.01, ### *P* < 0.001, significantly different from the value of normal group. * *P* < 0.05, ** *P* < 0.01, and *** *P* < 0.001 significantly different from the value of control group).

**Figure 5 nutrients-12-00926-f005:**
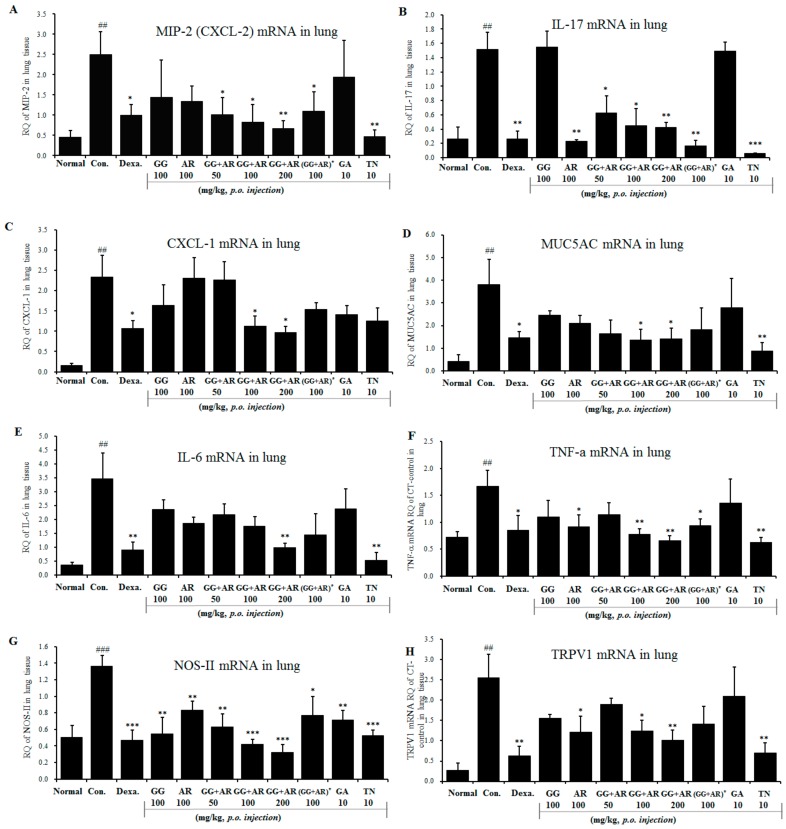
Effects of GG, AR, GG plus AR, GA, and TN on CXCL2, IL-17A, CXCL1, MUC5AC, IL-6, TNF-α NOS-II, and TRPV1 mRNA gene expression in the lung tissue of CFD-induced COPD model mice. CXCL2 (**A**), IL-17A (**B**), CXCL1 (**C**), MUC5AC (**D**), IL-6 (**E**), TNF-α (**F**), NOS-II (**G**), and TRPV1 (**H**) mRNA levels were measured in mouse lung tissue of each groups. Normal: normal group; Con.: CFD, CFD-induced COPD model group; Dexa.: CFD solution and administrated with dexamethasone 3 mg/kg group; GG 100: CFD solution and administrated with GG extract 100 mg/kg group; AR 100: CFD solution and administrated with AR extract 100 mg/kg group; GG + AR 50: CFD solution and administrated with GG plus AR extract 50 mg/kg group; GG + AR 100: CFD solution and administrated with GG plus AR extract 100 mg/kg group; GG + AR 200: CFD solution and administrated with GG plus AR extract 200 mg/kg group; GG + AR 100* (trial product): CFD solution and administrated with GG plus AR (trial product) 100 mg/kg group; GA 10: CFD solution and administrated with GA 10 mg/kg group; TN 10: CFD solution and administrated with TN 10 mg/kg group. The statistical significance of differences between control and treatment groups were assessed by ANOVA and Duncan’s multiple comparison test. Data represent the mean ± SEM of 8 independent experiments. (## *P* < 0.01, ### *P* < 0.001, significantly different from the value of normal group. * *P* < 0.05, ** *P* < 0.01, and *** *P* < 0.001 significantly different from the value of control group).

**Figure 6 nutrients-12-00926-f006:**
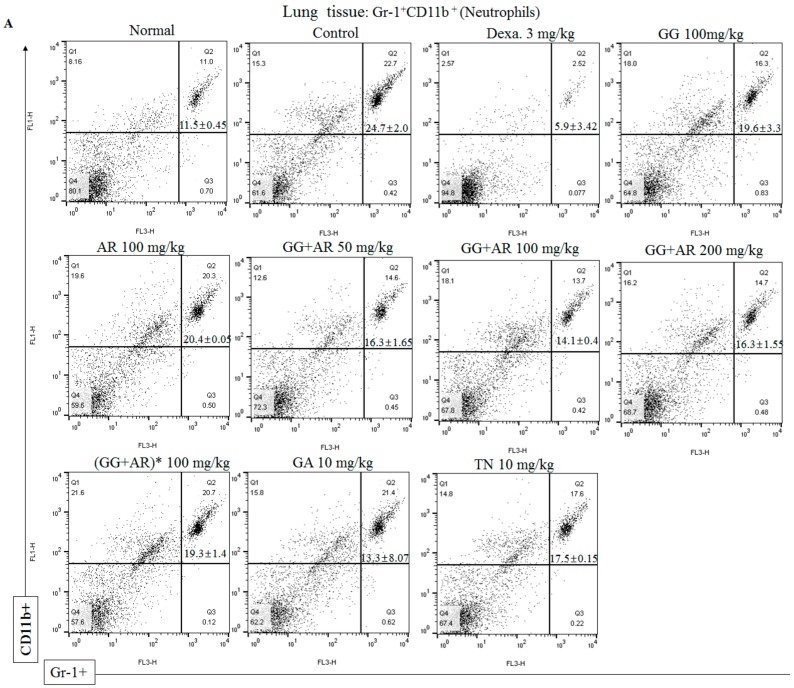
Inhibitory effect of GG, AR, GG plus AR, GA, and TN on CD11b+Gr-1+(high) neutrophils infiltration in lung tissue (**A**) and BALF (**B**) of COPD model mice. Samples were measured on flow cytometer and analyzed by two-color flow cytometry using a FACS Calibur device and CellQuest software (BD Biosciences, Mountain View, CA, USA). Absolute cell number of CD11b+Gr-1+(high) (neutrophils) present in lungs (**C**), and BALF (**D**). Normal: normal group; Con.: CFD, CFD-induced COPD model group; Dexa.: CFD solution and administrated with dexamethasone 3 mg/kg group; GG 100: CFD solution and administrated with GG extract 100 mg/kg group; AR 100: CFD solution and administrated with AR extract 100 mg/kg group; GG+AR 50: CFD solution and administrated with GG plus AR extract 50 mg/kg group; GG+AR 100: CFD solution and administrated with GG plus AR extract 100 mg/kg group; GG + AR 200: CFD solution and administrated with GG plus AR extract 200 mg/kg group; GG + AR 100* (trial product): CFD solution and administrated with GG plus AR (trial product) 100 mg/kg group; GA 10: CFD solution and administrated with GA 10 mg/kg group; TN 10: CFD solution and administrated with TN 10 mg/kg group. The statistical significance of differences between control and treatment groups were assessed by ANOVA and Duncan’s multiple comparison test. Data represent the mean ± SEM of 8 independent experiments. (## *P* < 0.01, significantly different from the value of normal group. * *P* < 0.05, ** *P* < 0.01, and *** *P* < 0.001 significantly different from the value of control group).

**Figure 7 nutrients-12-00926-f007:**
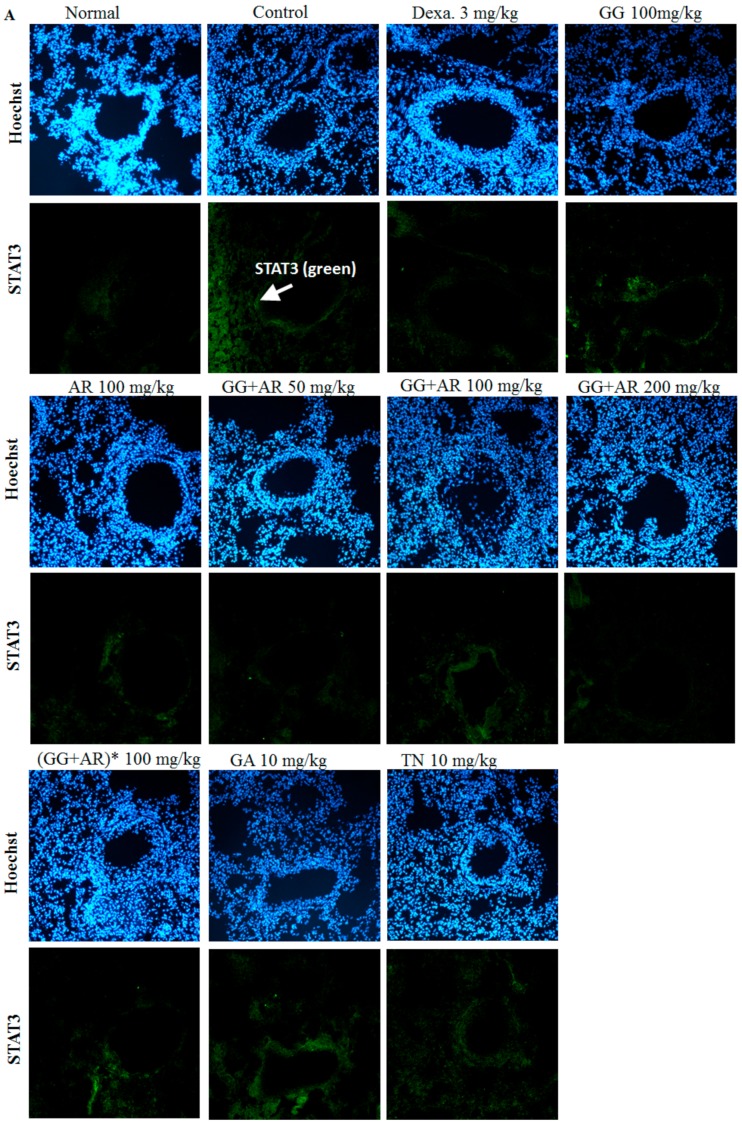
Inhibitory effect of GG, AR, GG plus AR, GA, and TN on STAT3 transcription factor using immunofluorescence analysis in lung tissues of COPD model mice. Mice were challenged by intratracheal injection of CFD, and then treated with GG, AR, GG plus AR, GA, and TN over a period of 12 days. The expression of STAT3 transcription factor in lung tissues was determined by immunofluorescence staining (**A**). The mean of fluorescence intensity quantified from images was obtained from 3 independent experiments using Image J software (**B**). Normal: normal group; Con.: CFD, CFD-induced COPD model group; Dexa.: CFD solution and administrated with dexamethasone 3 mg/kg group; GG 100: CFD solution and administrated with GG extract 100 mg/kg group; AR 100: CFD solution and administrated with AR extract 100 mg/kg group; GG+AR 50: CFD solution and administrated with GG plus AR extract 50 mg/kg group; GG + AR 100: CFD solution and administrated with GG plus AR extract 100 mg/kg group; GG + AR 200: CFD solution and administrated with GG plus AR extract 200 mg/kg group; GG + AR 100*(trial product): CFD solution and administrated with GG plus AR (trial product) 100 mg/kg group; GA 10: CFD solution and administrated with GA 10 mg/kg group; TN 10: CFD solution and administrated with TN 10 mg/kg group. The statistical significance of differences between control and treatment groups were assessed by ANOVA and Duncan’s multiple comparison test. Data represent the mean ± SEM of 8 independent experiments. (### *P* < 0.001, significantly different from the value of normal group. * *P* < 0.05 and ** *P* < 0.01 significantly different from the value of control group).

**Figure 8 nutrients-12-00926-f008:**
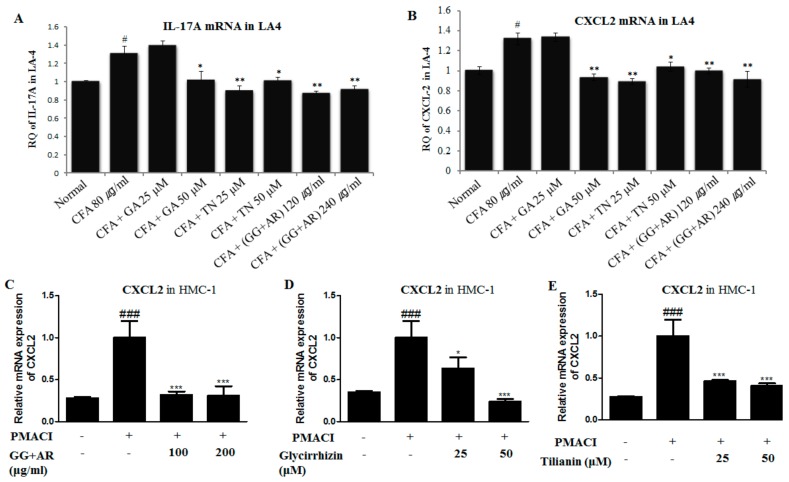
Inhibitory effect of GG plus AR, GA, and TN on CXCL2, IL-17A mRNA gene expression in vitro (LA4 and HMC-1 cell line). LA4 cells were pre-treated with of vehicle, GG plus AR (120 or 240 μg, GA, TN for 30 min prior to the addition of 80 ug/mL of CFA for 6 h. HMC-1 cells were pre-treated with of vehicle, GG plus AR (100 or 200 μg/mL), GA, TN for 30 min prior to the addition of 40 nM of PMA plus 1 μM of A23187 (PMACI) for 6 h. The cells were seeded 16 h before RNA preparation. IL-17A (**A**), CXCL2 (**B**) mRNA expression levels in LA4 and effects of GG plus AR (**C**), GA (**D**), and TN (**E**) on CXCL2 mRNA gene expression were measured using quantitative real-time PCR. The statistical significance of differences between control and treatment groups were assessed by ANOVA and Duncan’s multiple comparison test. The data shown represent mean ± SD of three independent experiments. # *P* < 0.05 and ### *P* < 0.001 vs. the normal control group, * *P* < 0.05, ** *P* < 0.01, and *** *P* < 0.001 vs. CFA-treated or the PMACI-treated group.

**Figure 9 nutrients-12-00926-f009:**
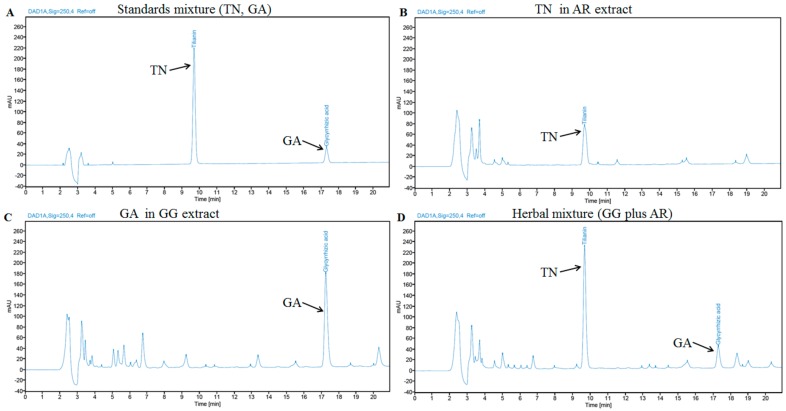
HPLC chromatogram of two standards mixture (**A**), AR extract (**B**), GG extract (**C**), and their mixture (**D**) at 250 nm. TN and GA were appeared at a retention time of approximately 9.67 and 17.28 min, respectively.

**Table 1 nutrients-12-00926-t001:** Primers and probe sequence used in real-time PCR analysis.

Gene	Primer	Oligonucleotide Sequence (5′-3′)
GAPDH	F	5′-CAATGAATACGGCTACAGCAAC-3′
	R	5′-AGGGAGATGCTCAGTGTTGG-3′
NOS-II	F	5′-CCCTTCCGAAGTTTCTGGCAGCAGC-3′
	R	5′-GGCTGTCAGAGCCTCGTGGCTTTGG-3′
MIP-2 (CXCL-2)	F	5′-ATGCCTGAAGACCCTGCCAAG-3′
	R	5′-GGTCAGTTAGCCTTGCCTTTG-3′
CXCL-1	F	5′-CCG AAG TCA TAG CCA CAC-3′
	R	5′-GTG CCA TCA GAG CAG TCT-3′
TNF-ɑ	F	5′-TTGACCTCAGCGCTGAGTTG-3′
	R	5′-CCTGTAGCCCACGTCGTAGC-3′
IL-6	F	5′-GTACTCCAGAAGACCAGAGG-3′
	R	5′-TGCTGGTGACAACCACGGCC-3′
IL-17A	F	5′-TCTCATCCAGCAAGAGATCC-3′
	R	5′-AGTTTGGGACCCCTTTACAC-3′
MUC5AC	F	5′-AGAATATCTTTCAGGACCCCTGCT-3′
	R	5′-ACACCAGTGCTGAGCATACTTTT-3′
TRPV1	F	5′-CATCTTCACCACGGCTGCTTAC-3′
	R	5′-CAGACAGGATCTCTCCAGTGAC-3′

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
