# Peer review of "Herbal Combinational Medication of *Glycyrrhiza glabra*, *Agastache rugosa* Containing Glycyrrhizic Acid, Tilianin Inhibits Neutrophilic Lung Inflammation by Affecting CXCL2, Interleukin-17/STAT3 Signal Pathways in a Murine Model of COPD"

_nutrients, 2020, doi:10.3390/nu12040926_

Round 1

Reviewer 1 Report

The manuscript presented for review is a study into possible application of a combination of herbal active ingredients as anti-inflammatory therapy in COPD.  The authors suggest that there are synergistic effects that may result when low dose herbal combination therapy is applied.  Overall the study is ambitious in nature and aims to investigate multiple permutations of Glycyrrhiza glabra, Agastache rugosa, and their active ingredients.  The search for novel therapeutics in COPD is of high value and this study is an important addition to the field.  In addition to this, the use of herbal compounds and their derivatives is of great interest to many in the medical community.  Thus, the value of this study is high, and the contribution can be great.

That being said, the manuscript itself is lacking cohesion.  The introduction and conclusion are overly descriptive and lack direction. The introduction itself is acceptable in the current form, however I suggest refinement of the conclusion.  The primary concern is that given the many permutations in this manuscript, it is difficult to follow what the authors are concluding.  Is the use of any combination of these compounds beneficial?  While it is clear that each compound has some effect on their model, there is no consensus on what combination or single active ingredient is most effective.  This should be addressed.  Do the authors think that their work produced a possible new therapeutic application?    

Results Section:     

Figure 2: Unclear what the total lung cell count is referencing.  Is this the total number of cells in their single cell suspension from a single mouse lung?  I believe this is the case, however this should be clarified in the methods and figure legend. There is no statistical designation on Figure 2C, however the authors mention changes in total cell number.  Was this just a typographical error or is there no significant difference in total lung cell number?   Did the authors see significant changes in the percent neutrophils in their cytospins, given the flow cytometry presented later it may be helpful to see this number?

Figure 3:  It would be helpful to see larger section of the lung histology, perhaps a stich of the entire lung at a higher magnification could be included for these conditions, if not there should be multiple images presented to give us an understanding of the overall lung histology.  The selection of a single field at high magnification does not fully demonstrate the significant differences between the groups suggested by the quantification in Fig 3D.  I understand that the paper could get very busy with the inclusion of so many different images, however perhaps these could be included in the supplementals and referenced.  A single representative section is not ideal for this analysis. Also, please include a distinction between # and * the figure legend.

Figure 4: Line 403, there is a typo, “treated with each samples”.  Please modify this.  Line 441-442.  It is unclear what the authors are trying to say with this statement.  Perhaps the language could be clarified here as well.

Figure 6:  My familiarity with this model of COPD is limited, however the number of neutrophils present in the BAL and whole lung samples is quite high.  As mentioned above it would be beneficial to see the percentages of these cells in the cytospins.  If these numbers correlate, then this concern would be decreased.  Overall the gating strategy is not clearly presented, did the authors simply label all cells with two colors and run this?  What QC was performed on the run prior to the collection of data on the GR1(high) and CD11b+ populations?  Please include this information in the methods. 

Figure 7:  The resolution of the IF images is poor.  It is difficult to see any STAT3 staining even in their positive control. Please address this. 

Finally, I am confused with the goal of figures 7 and 8.  Does this data add anything significant to their manuscript?  The authors have effectively presented the recruitment of neutrophils in their model, they have clearly demonstrated the effect of these compounds on the phenotype of their model and on the recruitment of immune cells.  The authors also very clearly demonstrate a change in gene expression of lung lysate.  However, there is no evidence presented to indicate that any specific cell type is driving this gene expression.  Thus, the in-vitro analysis of cell lines is not relevant for this model (Fig 8).  The work itself is enough without the in-vitro study.  If the authors do wish to add an in-vitro aspect, then I suggest the author’s first co-stain their histological section to identify the cells secreting their cytokine panel.  If that correlates with epithelial cells, then their in-vitro work is far more relevant.   

Author Response

Firstly, Reviewer #1 comments1:

Reviewer #1: That being said, the manuscript itself is lacking cohesion.  The introduction and conclusion are overly descriptive and lack direction. The introduction itself is acceptable in the current form, however I suggest refinement of the conclusion.  The primary concern is that given the many permutations in this manuscript, it is difficult to follow what the authors are concluding.  Is the use of any combination of these compounds beneficial?  While it is clear that each compound has some effect on their model, there is no consensus on what combination or single active ingredient is most effective.  This should be addressed.  Do the authors think that their work produced a possible new therapeutic application?

RESPONSE: ● First of all, thank you for your kind meaningful and helpful comment.

The text (in conclusion) has been revised(clarified or addressed more clearly) as suggested (as reviewer’s comment, “Refinement of the conclusion”) – page 28 in red sentence

Reviewer #1 comments2: Figure 2: Unclear what the total lung cell count is referencing.  Is this the total number of cells in their single cell suspension from a single mouse lung?  I believe this is the case, however this should be clarified in the methods and figure legend. There is no statistical designation on Figure 2C, however the authors mention changes in total cell number.  Was this just a typographical error or is there no significant difference in total lung cell number?   Did the authors see significant changes in the percent neutrophils in their cytospins, given the flow cytometry presented later it may be helpful to see this number?

RESPONSE: We revised "total cell number" into "total BALF cells" as shown in Fig.2B.

Mice were divided into 11 treatment groups (n=8 for each group)(described in materials and method)

 Single cell suspensions from lung tissues (each mouse lung) were isolated by mechanical disruption and total cell counts of each groups was calculated.(the value of mean, standard deviation and Standard errors.)  - page 4 in red sentence

  • This was addressed (clarified) in the methods and figure legend(. 2.4. Collection of bronchoalveolar lavage fluid (BALF) and lung cells section)
  • Although total lung cell number was changed(pattern results was shown), there is no significant difference in total lung cell number.
  • We see significant changes in the neutrophils average number in BAL cytospins (in Fig. 2 D). Also, flow cytometry presented later(Fig.6, although detection method is different), is helpful to see this number.

Reviewer #1 comments3:  It would be helpful to see larger section of the lung histology, perhaps a stich of the entire lung at a higher magnification could be included for these conditions, if not there should be multiple images presented to give us an understanding of the overall lung histology.  The selection of a single field at high magnification does not fully demonstrate the significant differences between the groups suggested by the quantification in Fig 3D.  I understand that the paper could get very busy with the inclusion of so many different images, however perhaps these could be included in the supplementals and referenced.  A single representative section is not ideal for this analysis. Also, please include a distinction between # and * the figure legend.

RESPONSE: * The text (image) has been revised(more larger section of the lung histology) as suggested (as reviewer’s comment) as possible.(so many groups, total image scale limited)

We condensed or truncated as much as we can express more clear conditions and overall lung histology .

- page 11-13, 14

Distinction between # and * the figure legend was added in the text(each figure legends) -- page 10,15, 16, 18, 21,23, 24 in red sentence

Reviewer #1 comments4:  Line 403, there is a typo, “treated with each samples”.  Please modify this.  Line 441-442.  It is unclear what the authors are trying to say with this statement.  Perhaps the language could be clarified here as well.

RESPONSE: * The text (in conclusion) has been revised as “treated with GG, AR, GG plus AR, GA and TN” (more clearly as reviewer’s comment)

Line 441-442. We think it is unnecessary (we agree with the reviewer’s comment in principle ), so this statement was deleted. - page 16 in red sentence

Reviewer #1 comments5:  Figure 6:  My familiarity with this model of COPD is limited, however the number of neutrophils present in the BAL and whole lung samples is quite high.  As mentioned above it would be beneficial to see the percentages of these cells in the cytospins.  If these numbers correlate, then this concern would be decreased.  Overall the gating strategy is not clearly presented, did the authors simply label all cells with two colors and run this?  What QC was performed on the run prior to the collection of data on the GR1(high) and CD11b+ populations?  Please include this information in the methods. 

RESPONSE: * We agree with the reviewer’s comment in principle. (revised in page 4, 2.5 section)

Although it is not sufficient expression, we already showed significant changes in the neutrophils average number in BAL cytospins (in Fig. 2 D) and percentages of GR1(high) and CD11b+(double positive cells) in BALF and lung(Fig. 6A,B, right upper Q2 part). Moreover, we think that it is more important to express the cell counts (absolute number) as shown in Fig.6C, 6D. In my opinion, percentage of cells shows the general pattern of cell proportion.

Generally, we used two-color (double staining with CD11b, Gr-1 described in materials and methods) FACS analysis in this experiments.

By standard methods recommended by the Becton-Dickinson, two color flow cytometry analysis was performed with scatter gates set on the lymphocyte fraction by forward and side scatter (SCC) and PE or FITC fluorescence FL2 with laser excitation at 488 nm.

Dead cells were excluded from analysis by appropriate gating strategies and propidium iodide (PI) staining.

The number of immunofluorescence-positive cells(double positive cells) was determined out of 5000 cells analyzed. A cell gate containing lymphocytes was established on the basis of forward and side light scatter. For determination of the borderline between stained and unstained cells, cells were also stained with mouse FITC-conjugated CD11b and PE-conjugated Gr-1. Percentages were calculated on the basis of the number of lymphocytes found in each quadrant.

For QC, multi-color analysis and/or quantitation was performed, then the FITC/PE Compensation Standard™ was performed on the run prior to collection of data on the Gr-1 and CD11b population same day.

Reviewer #1 comments6:  The resolution of the IF images is poor.  It is difficult to see any STAT3 staining even in their positive control. Please address this. 

Finally, I am confused with the goal of figures 7 and 8.  Does this data add anything significant to their manuscript?  The authors have effectively presented the recruitment of neutrophils in their model, they have clearly demonstrated the effect of these compounds on the phenotype of their model and on the recruitment of immune cells.  The authors also very clearly demonstrate a change in gene expression of lung lysate.  However, there is no evidence presented to indicate that any specific cell type is driving this gene expression.  Thus, the in-vitro analysis of cell lines is not relevant for this model (Fig 8).  The work itself is enough without the in-vitro study.  If the authors do wish to add an in-vitro aspect, then I suggest the author’s first co-stain their histological section to identify the cells secreting their cytokine panel.  If that correlates with epithelial cells, then their in-vitro work is far more relevant.   

RESPONSE: * We agree with the reviewer’s comment in principle .

** The text (image) has been revised(more larger section of the image) as suggested (as reviewer’s comment) as possible

** To explain the above significant aspects, we added the explanation in the discussion briefly(P27, in red sentence),.

Although it is not sufficient resolution(for its including so many images), we think it can be distinguished to some extent by differences (changes).

In our manuscript, we focused neutrophil infiltration, IL-17 produciton and STAT3 signal COPD model.

Although, it is not an direct link , our results(fig. 7,8 data) presents significant supports STAT3 signal (in vivo), and IL-17 production(gene expression) in other cell types(epithelial and mast cells which are main sources of these cytokines and chemokines by drug treatments.

As shown in discussuion, activation of STAT3 also induces the expression of another related nuclear receptor, RORα that synergizes with RORγt in Th17 cell differentiation [54]. Consistent with our findings, a recent study reported that STAT3 is activated in the lungs obtained from patients suffering from severe COPD. Activation of the STAT3 pathway is critical for persistent inflammation in lung tissues [60].

To explain the above significant aspects, we added the explanation in the discussion briefly,.

Ablation of STAT3 specifically in T cells resulted in a failure of TH17 polarization and reduced IL-17 production . An increasing body of evidence shows the requirement of STAT3 for Th17 cell development. Indeed, in mice STAT3 is absolutely required for the induction of IL-17, IL-17F, and RORgt (61)

61.Harris TJ, et al. Cutting edge: an in vivo requirement for STAT3 signaling in TH17 development and TH17-dependent autoimmunity. J Immunol 2007;179:4313–4317.

Although it is not sufficient direct data and our results are not strongly indicated in vivo, we proved the significant(meaningful) evidence concerning them (between herbal combination including major compounds and IL-17, CXCL-2 expression).   

Reviewer 2 Report

In this MS, Kim et al have investigated the beneficial effect of herbal combination - Glycyrrhiza glabra and Agastache rugosa in the invitro and invivo model of COPD. In this didactic and exhaustive MS, authors have used various molecular and imaging tools to provide a mechanistic insights against the pharmacological benefits of this herbal composition, which is the merit, while the linguistic or grammar errors observed all over the MS, dampens the enthusiasm. Here are the important comments - 

  1. Although it is trivial, it is the most important. Grammar and linguistic error could be noticed all over the MS, especially, the word "was" or "were" were missing in many places. Apart from these 2 words, many other errors were also noticed.
  2. In page 2, line 92 and 93 are highly confusing. A rewrite is required. Similarly, the line 96 in Page 3, too.
  3. Why did the authors not try evaluating the effect of this herbal composition in wild type animal in the absence of CFD, for a toxic effect? Did they measure the body weight? Authors are required to showcase the same.
  4. This MS mainly focusses on the Immuno effect of the herbal composition, while the quantification of airway remodeling (functional outcome of COPD) has been excluded. Authors are required to quantify the small airway remodeling and add the data in the Fig 3
  5. Authors should arrive at a strategy to condense or truncate the images in Fig 3. 
  6. Authors have used SPSS version 14, which is more than a decade old software. A newer version is recommended.
  7. Scale is missing Fig 2. Authors are recommended to improve the image quality of Fig 2 and also recommended to provide higher magnification.
  8. Authors are required to highlight the pathological features in Fig 3A - 3C in the control section using arrow points. 
  9. As a proof of concept, authors are required to provide western blot confirmation for the STAT3 expression in the Fig 7. Further, they should improve the image quality, especially for STAT3 staining.

Author Response

Reviewer #2 comments1: Although it is trivial, it is the most important. Grammar and linguistic error could be noticed all over the MS, especially, the word "was" or "were" were missing in many places. Apart from these 2 words, many other errors were also noticed.

RESPONSE: ● Thank you for your kind meaningful and helpful comment.

The text was revised with correct language and better descriptions (or checked again) as possible. (English correction has been made in www.editage.co.kr- english-edit)

Reviewer #2 comments2: In page 2, line 92 and 93 are highly confusing. A rewrite is required. Similarly, the line 96 in Page 3, too.

RESPONSE: ● The text has been revised as suggested (rewritten as reviewer’s comment)- page 2~3 in red sentence

Reviewer #2 comments3 : Why did the authors not try evaluating the effect of this herbal composition in wild type animal in the absence of CFD, for a toxic effect? Did they measure the body weight? Authors are required to showcase the same.

RESPONSE; We agree with the reviewer’s comment in principle.

In a number of previously published papers, slight therapeutic activities or no significant effects of potential drugs were shown in numerous wild types compared to diverse disease models. Therefore, we thought that it is more important to evaluate the effects of potential drugs in disease groups at safe and effective dose range.

As mentioned in discussion(reference 32-34 and below references), numerous studies were accomplished in DEP, coal fly ash (at non-toxic dose range) induced lung injury model rather than wild types.

Several investigators have used more complex chemical mixtures, such as diesel exhaust particles (DEP) or coal fly ash (CFA), which contribute to the complex make-up of urban collected PM.

In many other reports listed below, CFA, DEP(Standard reference material 2975 (SRM2975)) were used to induce lung injury. We used(selected) representative doses(selected minimal doses as safe and effective range) to induce lung injury(COPD model in our case) based on reviewed related studies as belows.

We administered 100 μL of CFD (Coal 5 mg/mL, Fly ash 10 mg/mL, and diesel-exhaust particles (DEP) 5 mg/mL) in saline by intratracheal instillation thrice at 3-day intervals for 12 days(mentioned in materials and methods). (In 100 μL of CFD include about CFD 2 mg/mL(absolute concentration))

▶ Aerosolized CFA in a nose-only exposure system for 4 h/day for 3 days and examined 18 or 36 h after the last exposure to CFA. Average concentration of CFA in the PM2.5 range was 1400 ± 150 ug/m3 , of which 600 ± 70 ug/m3 was in the PM1 range. Toxicol Sci. 2006 Oct;93(2):390-9.

▶ Pulmonary toxicology studies using CFA particles have generally shown few effects following instillation of 2–10 mg/kg in rats (Borm, 1997). Ann. Occup. Hyg. 41, 659–676.

▶ The CP(candle light combustion particles) were administered as a low (LCP; 0.5 mg/kg) and high (HCP; 5 mg/kg)dose by intratracheal (i.t.) instillation. The A-DEP and SRM2975 groupsreceived a dose of 5 mg/kg. Toxicol Lett. 2017 Jul 5;276:31-38. 

▶ CFA (CFA11 and CFA16) particles that wereinstilled intratracheally with 100μg of COAL. Sci Total Environ. 2018 Jun 1;625:589-599. 

▶exposed to coal ash inhalation (10 mg/m3) for 3 h per day totaling 30 days of coal ash circulation. Food and Chemical Toxicology 133 (2019) 110766.

▶ exposed to either clean air or 20 mg/m3 DEP (Standard reference material 2975 (SRM2975), Toxicolog. 2009 Oct 1;264(1-2):61-8. 

▶ BALB/c mice were exposed intratracheally (i.t.) to 50 μL of saline, DEP (150 μg), and/or HDM (10 μg) 3 times a week over a 3-week period. Several investigators report significant lung inflammation in response to airway exposure to DEP and fly ash (Brandt et al., 2015). Journal of Allergy and Clinical Immunology. Volume 136, Issue 2, August 2015, Pages 295-303.e7

Moreover, DEP (Standard reference material 2975 (SRM2975) was used in numerous researches(in safe range doses according to reference guidelines) and so on.

Above studies showed that total number of neutrophils in bronchoalveolar lavage fluid (BALF) following exposure to CFA was significantly increased along with significantly elevated blood neutrophils. Exposure to CFA caused slight increases in macrophage inflammatory protein-2, and marked increases in transferrin in BALF.

We focused on thereapeutic effects of samples in disease models. Unfortunately, we did not try evaluating the effect of the herbal composision in wild type.

Previously(in preliminary experiment), we used the protocol described in materials and methods. And we compared normal group to control group, and control group to experimental group.

In our study, there were non-toxic effects according to the hematological parameters in our experimental groups (at used concentration).

Reviewer #2 comments4 : This MS mainly focusses on the Immuno effect of the herbal composition, while the quantification of airway remodeling (functional outcome of COPD) has been excluded. Authors are required to quantify the small airway remodeling and add the data in the Fig 3

RESPONSE; We think that Reviewer’s comment was thought to be important valid point.

Unfortunately, quantification of airway remodeling (functional outcome of COPD) has been excluded(not detected).

** Generally, in a murine model of COPD, histopathological analysis was accomplished instead of functional outcome of COPD(FEV1, Cumputed tomography) (Perhaps, mouse is too small to evaluate functional test)

  • In mouse model, histopathological analysis, Journal of Ethnopharmacology 217 (2018) 152–162
  • In mouse model, Lung Histological Analysis, 2020 Feb 28;12(3). pii: E657.
  • In mouse model, histological analysis, Inflamm Res.2020 Mar 6. doi: 10.1007/s00011-020-01333-1
  • In mouse model, histopathological examination, Am J Physiol Lung Cell Mol Physiol.2020 Mar 4. doi: 10.1152/ajplung.00214.2019.
  • In mouse model, Histological analysis of the lung , Environmental Research 161 (2018) 304–313
  • In mouse model., Histology and Immunohistochemistry. Evidence-Based Complementary and Alternative Medicine. Volume 2012, Article ID 769830, 10 pages
  • In mouse model, Histopathology Examination, Inflammation, Vol. 40, No. 3, June 2017.
  • In mouse model, Lung Tissue Histopathology, Front Pharmacol.2018 Sep 21;9:1064.
  • In mouse model, Lung tissue histopathology, Phytomedicine 59 (2019) 152777.
  • In mouse model, Histopathological analyzes, Biomedicine & Pharmacotherapy 99 (2018) 591–597.
  • In mouse model, Histological Examination, Evidence-Based Complementary and Alternative Medicine Volume 2018, Article ID 4265790, 13

and so on.

** In a rat model of COPD, pulmonary function was detected with a Pulmonary Function Testing (PFT) system (DSI, Buxco, USA). Functional residual capacity (FRC), forced expiratory volume in 100 ms (FEV100), forced vital capacity (FVC), maximum midexpiratory flow (MMEF), and peak expiratory flow (PEF) etc.

  • In rat model, pulmonary function test(FEV100, FVC, etc.), Biomedicine & Pharmacotherapy 123 (2020) 109735
  • In rat model, Setup of the cough recording system, Ther Adv Respir Dis. 2019 Jan-Dec;13:1753466619877960. 
  • In rat model, quantitatively evaluate the gas exchange (by MRI), NMR Biomed.2018 Sep;31(9):e3961.
  • In rat model, Pulmonary Function Measurement, Mediators Inflamm. 2016; 2016: 4192483.
  • In rat model, The respiratory functional parameters and lung capacity , Eur J Pharmacol. 2015 May 15;755:88-94.
  • In rat model, lung function test, Int J Clin Exp Pathol.2014 Dec 1;7(12):8553-62. 
  • In rat model, total lung volume, Acad Radiol.2010 Nov;17(11):1433-43.
  • In rat model, measurement of secretary activity, J Occup Med Toxicol.2006 Jun 7;1:12.
  • In rat model, pulmonaryarterial pressure, Am J Respir Crit Care Med. 2005 Oct 15;172(8):987-93.

and so on.

The airway structure involving the trachea and bronchi, so we detected in histopathological analysis in trachea (already investigated).We measured AB/PAS-positive areas and the trachea (including epithelial area). This result was added in the Fig.3 (Fig.3D).

Although, it is not an direct link (associated with COPD and airway remodeling), we detected TRPV1 gene (Cough related gene) expression in lung instead of airway remodeling and explained in the discussion section. (p25, TRPV1 plays a crucial role in the cough reflex. TRPV1 are significantly related to cough [51].)

As mentioned above, and reviewer’s comment, we focused on the immune effects of herbal composition.

As reviewer’s meaningful and helpful comment, further investigation about the quantification of airway remodeling (functional outcome of COPD) and complicated mechanism should be accomplished in separate study focused on this theme in the future study.

Reviewer #2 comments5 : Authors should arrive at a strategy to condense or truncate the images in Fig 3. 

RESPONSE: ● The text has been revised as suggested (as reviewer’s comment) as possible.

As reviewer 1 suggested(Similar concern), the image has been revised(more larger section of the lung histology).

  • Thank you for your kind meaningful and helpful comment.

Reviewer #2 comments6 : Authors have used SPSS version 14, which is more than a decade old software. A newer version is recommended.

RESPONSE: ● The text has been revised as suggested (as reviewer’s comment) another software in “ Prism v. 7.0 (GraphPad Inc., San Diego, CA)”.  – Page 7, in red sentence

We re-analyzed the statistics as was suggested (as reviewer’s comment).  

P-values showed a slight difference, but statistical significance did not change with the exception of a few case that is not already represented.

Reviewer #2 comments7 : Scale is missing Fig 2. Authors are recommended to improve the image quality of Fig 2 and also recommended to provide higher magnification.

RESPONSE: ● The text has been revised as suggested (as reviewer’s comment) as possible. (Fig.2)

Scale checked and improved the image quality(as possible with in text limits)

 Reviewer #2 comments8 : Authors are required to highlight the pathological features in Fig 3A - 3C in the control section using arrow points. 

RESPONSE: ● The text (in Fig3A,-3C control section)has been revised as suggested (as reviewer’s comment) as possible.

Reviewer #2 comments9 : As a proof of concept, authors are required to provide western blot confirmation for the STAT3 expression in the Fig 7. Further, they should improve the image quality, especially for STAT3 staining.

RESPONSE: ● We think that Reviewer’s comment was thought to be important valid point.

** The text (image) has been revised(more larger section of the image) as suggested (as reviewer’s comment) as possible. (Improved the image)

** To explain and support the above significant aspects, we added the explanation(related references) in the discussion briefly(P27, in red sentence, p31 references)- An related link (evidence) associated with our herbal compositions and STAT3.

  1. Gu, M.Y.;Chun, Y.S.;Zhao, D.; Ryu, S.Y.; Yang, H.O. Glycyrrhiza uralensis and Semilicoisoflavone B Reduce Aβ Secretion by Increasing PPARγ Expression and Inhibiting STAT3 Phosphorylation to Inhibit BACE1 Expression. Mol. Nutr. Food Res2018, 62, e1700633.
  2. Menegazzi, M.;Di-Paola, R.;Mazzon, E.; Genovese, T.; Crisafulli, C.; Dal-Bosco, M.; Zou, Z.; Suzuki, H.; Cuzzocrea, S. Glycyrrhizin attenuates the development of carrageenan-induced lung injury in mice. Pharmacol Res2008, 58, 22-31.
  3. Jiang, Y.X.;Dai, Y.Y.;Pan, Y.F.; Wu, X.M.; Yang, Y.; Bian, K.; Zhang, D.D. Total Flavonoids from Radix Glycyrrhiza Exert Anti-Inflammatory and Antitumorigenic Effects by Inactivating iNOS Signaling Pathways. Evid. Based Complement. Alternat. Med2018, 2018, 6714282.
  4. Cui, B.W.;Bai, T.;Yang, Y.; Zhang, Y.; Jiang, M.; Yang, H.X.; Wu, M.; Liu, J.; Qiao, C.Y.; Zhan, Z.Y.; et al. Thymoquinone Attenuates Acetaminophen Overdose-Induced Acute Liver Injury and Inflammation Via Regulation of JNK and AMPK Signaling Pathway. Am J Chin Med2019, 47, 577-594.

Unfortunately, western blot confirmation for the STAT3 expression in the Fig 7.(lung tissues in vivo model)  has been excluded(not detected), because lung tissues are not sufficient for investigation of diverse targets(including numerous signal targets), experiments. Lung tissues in our study model (COPD) are used diversely to investigate RNA expression, IHF detection(numerous targets are detected including targets which is not significantly changed), histopathological analysis and cell counts.  For western analysis, appropriate study design have to be made (selection of method, targets and so on). 

Although it is not sufficient direct data about stat3 expression in vivo model and our results are not strongly indicated stat3 signal.

Regardless, we think that the data presented is extensive and experimentally sound and we proved the significant(meaningful) evidence concerning them (between herbal combination including major compounds and IL-17, STAT3, CXCL-2 expression).   

Also, as reviewer’s meaningful and helpful comment, future research about western blot confirmation for the some protein expression of lung tissues in vivo model and complicated mechanism should be considered and accomplished in separate investigation focused on this theme in the future study.

Round 2

Reviewer 1 Report

The authors have submitted a revised version of their original manuscript exploring the possible application of an herbal combination therapy in COPD.  In general, the authors have addressed some of the concerns with the original manuscript, however there are several concerns remaining.  Again, I want to emphasize the value of the study and the ambitious nature of the work.  This study can be an excellent addition to the field. 

One primary issue with the original submission that continues to be present in the revision is the lack of cohesion in the introduction and discussion.  The overly descriptive nature of both sections makes reading of the sections rather laborious.  This is a minor concern overall, but the authors are encouraged to carefully evaluate the sections and shorten both to include the most relevant ideas in a clear and concise manner. 

Figure 1: It appears that the figure has been broken into two parts.  Subfigure 1C appears after the legend.  Please correct this so that the panels are all before the legend.

Figure 3:  The representative sections are incongruent with the quantification presented.  For example, in the H&E the authors score GG 100 mg/kg as an inflammatory score of 1 and TN 10mg/kg as the same inflammatory score of 1.  I would not agree that the two representative images have the same degree of cell infiltration.  This may be a result of the scoring system, 0-2 is a very narrow range.  The same is true for the collage deposition in the MT panels of GG and TN.  In these sections there is more collagen deposition in the TN section, but the TN section is scored as a 0 for collagen and the GG section is scored as a 1.  This may simply be a result of the image selected as the representative image, overall the lung may in fact be more congruent with the scores.  There are some possible solutions.  The first would be an inclusion of multiple images taken throughout the slide demonstrating, for example, that there is more collagen deposition in the GG 100 mg/kg lung than in the TN 10mg/kg treatment.  A second way to solve this would be to include a quantification using picrosirius red.  Finally, a third way to confirm this would be to do a sircol assay.  Depending on the desire of the authors to perform more experiments, it seems most prudent to simply include more images that support the scoring system.  I would also suggest adding these as supplemental images not in the main body of the manuscript.  I also emphasize that this is just an example of the discrepancy observed in the scoring of figure 1, all the images should be checked to ensure accuracy.

The authors included a blue arrow in the positive control highlighting the pathology that was quantified, these blue arrows should be included in all images not just the control group.  Rather than including the text directly in the images, the text should be added either to the legend or just below each sub figure to prevent the addition of extraneous text over the images captured.

Figure 7:  The quality of the STAT3 fluorescent images makes interpretation practically impossible.  For example, the authors quantify the TM 10mg/kg image with a mean fluorescence intensity of 80, however this image is completely black with a full absence of green staining.  In fact, the control case scored at a mean intensity of 40, has more visible green fluorescence.  The authors suggest that the resolution is an issue due to the number of sections included, however even zoomed in it is not possible to see any positive staining in their TM 10mg/kg section.  Overall if it is not possible to improve on this, the figure should be removed from the manuscript. 

Figure 8:  The authors suggest that paired with the observations reported in figure 7, it is relevant to include the in-vitro work in figure 8.  While I do not share this opinion, I do appreciate the added explanation in the discussion giving credence to the inclusion of this figure.  If the authors are unable to improve on the quality of figure 7, then I would strongly suggest the authors remove this figure from the main body of the manuscript as well.                                   

Discussion:  From the point of view of this reviewer it appears that Tilianin alone is enough to induce much of the observed inhibition in COPD pathology.  The authors suggest that GG and AR in combination is effective and that there may be some synergistic effects of the two.  Please comment on how the two in combination are an improvement over Tilianin alone within the context of this study.  Finally, please clarify the statement “GG and AR in the ratio of 1:4 by weight”, which of their doses does this correspond to 50, 100, or 200 mg/kg?   

Author Response

Reviewer #1 comment1: Figure 1: It appears that the figure has been broken into two parts.  Subfigure 1C appears after the legend.  Please correct this so that the panels are all before the legend.

RESPONSE: ● First of all, thank you for your kind helpful comment.

Reviewer commented that subfigure 1c appears after the legend. In Figure1, subfigure 1C panel is located before the legend(revised).

Comment2: Figure 3:  The representative sections are incongruent with the quantification presented.  For example, in the H&E the authors score GG 100 mg/kg as an inflammatory score of 1 and TN 10mg/kg as the same inflammatory score of 1.  I would not agree that the two representative images have the same degree of cell infiltration.  This may be a result of the scoring system, 0-2 is a very narrow range.  The same is true for the collage deposition in the MT panels of GG and TN.  In these sections there is more collagen deposition in the TN section, but the TN section is scored as a 0 for collagen and the GG section is scored as a 1.  This may simply be a result of the image selected as the representative image, overall the lung may in fact be more congruent with the scores.  There are some possible solutions.  The first would be an inclusion of multiple images taken throughout the slide demonstrating, for example, that there is more collagen deposition in the GG 100 mg/kg lung than in the TN 10mg/kg treatment.  A second way to solve this would be to include a quantification using picrosirius red.  Finally, a third way to confirm this would be to do a sircol assay.  Depending on the desire of the authors to perform more experiments, it seems most prudent to simply include more images that support the scoring system.  I would also suggest adding these as supplemental images not in the main body of the manuscript.  I also emphasize that this is just an example of the discrepancy observed in the scoring of figure 1, all the images should be checked to ensure accuracy.

RESPONSE: ● First of all, thank you for your kind helpful comment.

We agree with the reviewer’s comment that this may be a result of the scoring system, 0-2 is a very narrow range. We checked again carefully original score data sheet. As the reviewer pointed out, in GG, TN treated sections, inflammatory score and collagen deposition score was revised based on original score(modified). Inflammatory scores of GG and TN are 1, 2 respectively, collagen deposition scores of GG and TN are 0, 1 respectively. Figure 3E image has been revised. (PAGE 14)

The authors included a blue arrow in the positive control highlighting the pathology that was quantified, these blue arrows should be included in all images not just the control group.  Rather than including the text directly in the images, the text should be added either to the legend or just below each sub figure to prevent the addition of extraneous text over the images captured.

The text (image) has been revised as suggested (as reviewer’s comment) as possible. Blue arrows was included in all images not just the control group.  The text was added either to the figure legend. (PAGE 15)

Comment3: Figure 7:  The quality of the STAT3 fluorescent images makes interpretation practically impossible.  For example, the authors quantify the TM 10mg/kg image with a mean fluorescence intensity of 80, however this image is completely black with a full absence of green staining.  In fact, the control case scored at a mean intensity of 40, has more visible green fluorescence.  The authors suggest that the resolution is an issue due to the number of sections included, however even zoomed in it is not possible to see any positive staining in their TM 10mg/kg section.  Overall if it is not possible to improve on this, the figure should be removed from the manuscript. 

RESPONSE: ● First of all, thank you for your kind helpful comment. We agree with the reviewer’s comment in principle.

Although we can see the positive stain of STAT in TN 10mg/kg representative image in detail or larger scale , we changed representative image (near the average value) of TN group into more visible one. Also, the control case scored at a mean intensity of 40, has been changed more representative(near the average value) one.  

Comment4: Figure 8:  The authors suggest that paired with the observations reported in figure 7, it is relevant to include the in-vitro work in figure 8.  While I do not share this opinion, I do appreciate the added explanation in the discussion giving credence to the inclusion of this figure.  If the authors are unable to improve on the quality of figure 7, then I would strongly suggest the authors remove this figure from the main body of the manuscript as well.     

RESPONSE: ● First of all, thank you for your kind helpful comment.

We agree with the reviewer’s comment in principle. We improved the quality of figure 7 as reviewer’s comment.

We explained the relationship between IL-17 production, CXCL2 production(Figure 4A,4B, 5A, 5B; in vivo results), STAT3 expression (Figure 7; in vivo), and IL-17 production, CXCL2 production(Figure 8A-E; in vitro results). In other words, Figure 8 is a verification in vitro test for Figure 4 and 7(especially figure 4).

As previously described in results (Figure 4A,4B, 5A, 5B; in vivo results) and discussion, we focused neutrophil infiltration, IL-17 production, CXCL2 production and STAT3 signal COPD model. Th17 cells release IL-17A that acts on airway epithelial cells to release CXCL1 and CXCL2 that in turn attracts neutrophils. Th17 cells also release IL-21 that promotes Th17-cell differentiation via the STAT3 transcription factor [29]. Indeed, in mice STAT3 is absolutely required for the induction of IL-17, IL-17F, and RORgt (61)

Although it is not a full explanation of their relevance, follow explanation was added in discussion for credence (p27) and their references are added [62-63]. (p31-32)

  • IL-17 is produced by inflammatory cells and targets structural cells such as epithelial cells [62]. Mast cells release neutrophil chemoattractants CXCL1 and CXCL2 in both granular and newly synthesized forms [63]. To confirm the underlying mechanisms of CXCL2 and IL-17A gene expression and neutrophil migration in vivo [Figure 4A,B, Figure 5A,B], we tested the mRNA gene-expression levels of CXCL2 and IL-17A in the mouse epithelial cell line LA-4 and human mast-cell lines which are the main sources of these cytokines.                

Comment5: Discussion:  From the point of view of this reviewer it appears that Tilianin alone is enough to induce much of the observed inhibition in COPD pathology.  The authors suggest that GG and AR in combination is effective and that there may be some synergistic effects of the two.  Please comment on how the two in combination are an improvement over Tilianin alone within the context of this study.  Finally, please clarify the statement “GG and AR in the ratio of 1:4 by weight”, which of their doses does this correspond to 50, 100, or 200 mg/kg?   

RESPONSE: First of all, thank you for your kind helpful comment.

* We agree with the reviewer’s comment to some extent. 

However, in Fig.3E(histopathological results), Fig.4B(IL-17 in BALF), Fig.5C(cxcl-1, NOS-2 absolute value), and especially STAT3 (Fig. 7B), GG plus AR combination is more effective than tilianin.

Also, most of the other results of GG plus AR combination and tilianin are expected to be similar to each other. We thought that tilianin alone is not enough to induce these effects.   

* Follow explanation was added in discussion (p28)

Probably, more herbal ingredients of the mixture of GG and AR mutually enhance each other’s effect more significantly than the simple sum of these ingredients or individual agents including tilianin and glycyrrhizic acid. The potential synergistic mechanisms of the mixture of GG and AR can be explained by enhancing each other’s activity, and interacting with multiple targets of a pathway and its crosstalk pathways. Herbal products contain plant extracts, which are complex mixtures of various compounds. As with traditional drugs, it is necessary to validate their efficacy and safety through preclinical at the molecular levels and clinical studies.

* The statement “Herbal combinational medication of GG and AR in the ratio of 1:4 by weight” was revised as “Herbal combinational medication of GG and AR (100 mg/kg) in the ratio 1:4 by weight”.